# Comparative epigenetic analysis of tumour initiating cells and syngeneic EPSC-derived neural stem cells in glioblastoma

Claire Vinel[1], Gabriel Rosser[1,9], Loredana Guglielmi[1,9], Myrianni Constantinou [1,9], Nicola Pomella[1],
Xinyu Zhang[1], James R. Boot[1], Tania A. Jones [1], Thomas O. Millner[1], Anaelle A. Dumas[1], Vardhman Rakyan[1],
Jeremy Rees[2], Jamie L. Thompson[3], Juho Vuononvirta [4], Suchita Nadkarni[4], Tedani El Assan[2], Natasha Aley[5],
Yung-Yao Lin [1,6], Pentao Liu [7], Sven Nelander [8], Denise Sheer [1], Catherine L. R. Merry [3],
Federica Marelli-Berg [4], Sebastian Brandner [2,5] & Silvia Marino [1✉]

Epigenetic mechanisms which play an essential role in normal developmental processes, such as self-renewal and fate specification of neural stem cells (NSC) are also responsible for some of the changes in the glioblastoma (GBM) genome. Here we develop a strategy to compare the epigenetic and transcriptional make-up of primary GBM cells (GIC) with patient-matched expanded potential stem cell (EPSC)-derived NSC (iNSC). Using a comparative analysis of the transcriptome of syngeneic GIC/iNSC pairs, we identify a glycosaminoglycan (GAG)-mediated mechanism of recruitment of regulatory T cells (Tregs) in GBM. Integrated analysis of the transcriptome and DNA methylome of GBM cells identifies druggable target genes and patient-specific prediction of drug response in primary GIC cultures, which is validated in 3D and in vivo models. Taken together, we provide a proof of principle that this experimental pipeline has the potential to identify patient-specific disease mechanisms and druggable targets in GBM.

[1] Blizard Institute, Barts and The London School of Medicine and Dentistry, Queen Mary University London, London, UK. [2] Division of Neuropathology, The National Hospital for Neurology and Neurosurgery, University College London Hospitals NHS Foundation Trust, Queen Square, London, UK. [3] Stem Cell Glycobiology Group, Biodiscovery Institute, University of Nottingham, Nottingham, UK. [4] The William Harvey Research Institute, Barts and The London School of Medicine and Dentistry, Queen Mary University London, London, UK. [5] Department of Neurodegenerative Disease, UCL Queen Square Institute of Neurology, Queen Square, London, UK. [6] Stem Cell Laboratory, National Bowel Research Centre, Barts and the London School of Medicine and Dentistry, Queen Mary University of London, 2 Newark Street, London, UK. [7] Faculty of Medicine, School of Biomedical Sciences, The University of Hong Kong, Hong Kong, Hong Kong. [8] Department of Immunology Genetics and Pathology, Uppsala University, Uppsala, Sweden. [9] These authors contributed equally: Gabriel Rosser, Loredana Guglielmi, Myrianni Constantinou. ✉email: s.marino@qmul.ac.uk

Glioblastoma (GBM) is the most common adult primary brain cancer; it cannot be completely resected by surgery and is resistant to conventional anticancer treatments, with a median survival of <15 months[1]. Significant advances in understanding the genetics of these tumours[2,3] have had, so far, only a modest impact on the therapeutic approach to the disease.

One of the main challenges in effectively tackling GBM is a remarkable heterogeneity at both intertumoural and intratumoural levels. Defined genetic alterations together with DNA methylation and transcriptional profiles have identified molecular subtypes which account for the intertumoural heterogeneity of GBM[4,5]. The intratumoural heterogeneity though is determined not only by the presence of non-neoplastic cells, such as immune cells, brain cells, and vessels within the tumour ecosystem but also by substantial molecular and functional differences, as well as plasticity within the tumour cell population itself (recently reviewed in ref. [6]). Cancer stem cells (CSC) have been shown to contribute significantly to the tumour component of the intratumoural heterogeneity and therefore to resistance to therapy and tumour recurrence in many cancers (recently reviewed in ref. [7]). In GBM, CSC (called GBM stem cells (GSC) or GBM initiating cells (GIC)), exhibit properties of self-renewal and multilineage differentiation, similarly to neural stem cells (NSC) in brain development and homeostasis, and give rise to neoplasms recapitulating the tumour from which they were derived in orthotopic xenograft models[8]. Single-cell transcriptional profiling of GBM has recently confirmed that GIC recapitulates a normal neurodevelopmental hierarchy with glial progenitor-like cells at its apex and that they contribute to drug resistance[9]. The molecular and functional relationship between GIC and NSC is further corroborated by data in genetically engineered mouse models[10,11] and human tissue samples[12] providing compelling evidence that they are a cell of origin of glial tumours, including GBM. The isolation, enrichment, and propagation of GIC in vitro either as adherent cultures or neurospheres is a powerful tool for screening and functional validation of targets. These primary patient-derived cell lines retain the ability to generate tumours reflecting the heterogeneity of the parental tumour as shown in barcoding experiments followed by characterization of clonal expansion[13]. They recapitulate to some degree the cellular states and plasticity found in GBM[14] and core regulatory circuits identified in these cells are maintained in matched primary tumours[15]. Epigenetic remodelling which locks GIC into an undifferentiated state upon prolonged culturing[9,16] and the lack of interactions between these cells and various other cell types existing in vivo are known limitations of this experimental system.

Characterization of the molecular landscape of gliomas has highlighted a key role for epigenetic deregulation in these tumours. Methylation of the promoter of the DNA repair gene MGMT was the first example of an epigenetic regulation of the expression of a gene serving as a predictive biomarker of drug response in neuro-oncology[17]. At a global DNA level, the methylome of brain tumours was recently utilized to develop a machine learning approach for a DNA methylation-based classification of brain tumours which enables standardization and improved precision of tumour diagnoses[18]. Harnessing the expanding knowledge of epigenetic deregulation in brain tumours, particularly in those currently untreatable, to identify novel therapeutic approaches is an ambitious but now a realistic opportunity.

Characterization of epigenetic deregulation in GBM has so far relied on comparisons between tumour samples obtained from different patients[4,19] or between GBM and progenitor cells derived from an unrelated donor[15,20] or foetal brain[9]. Here, we have harnessed state-of-the-art stem cell technologies and next-generation sequencing methods to compare the epigenetic and transcriptional profile of IDH-wildtype GBM with that of patient-matched normal expanded potential stem cell (EPSC)-derived NSC (iNSC). We show that syngeneic iNSC can be used as a proxy to identify disease-specific mechanisms and to predict drug response in in vitro and in vivo systems.

## Results

### Establishment and characterization of patient-derived GIC and syngeneic iNSC lines.
We established an experimental pipeline to derive GIC and patient-matched (syngeneic) iNSC from 10 patients with IDH-wildtype GBM (Fig. 1a and Table S1 for clinical and molecular data). hGIC lines were established according to standard protocols[21,22]. Cases were included in the study where the molecular subgrouping, determined by DNA methylation arrays[4] was conserved between bulk tumour and hGIC (Fig. 1b) and where correlation analysis of gene expression revealed a good normalized correlation value between samples from the same patient (Fig. S1a). A match of copy number alterations between bulk tumour and corresponding GIC was confirmed by visual inspection of whole genome and chromosome-level plots (Fig. S1b). The tumour-initiating potential of all 10 GICs was confirmed in vivo by an intracerebral xenografting assay in NOD SCID mice (Fig. S1c).

Syngeneic fibroblast cultures (Fig. 1a and Fig. S2a) were derived from the dura mater of the same patient and reprogrammed to expanded potential stem cells (EPSC)[23]. Selected lines were characterized by immunostaining for OCT3/4, SOX2, NANOG, SSEA4 and TRA-1-60 (Fig. 1c, top). Hierarchical clustering of DNA methylation array data demonstrated that EPSC cluster with the reference "embryonic stem cells" (ESC) (Fig. S2c, d) from several different studies[24–26]. To exclude significant epigenetic aberrations, which might have introduced bias into downstream analyses of the EPSC or cell lines derived therefrom, we studied the methylome and transcriptome of our EPSC relative to those of ESC lines in greater detail. We show that the number of differentially methylated regions (DMRs) detected between the EPSC lines and two reference ESC lines[27,28] is similar to the number observed with published iPSC lines generated using different strategies (HipSci Initiative and[28,29]) (Fig. S2e). This difference in the methylome is small in comparison to the 321 DMRs identified when the two reference ESC lines are compared with each other. Very limited overlap of DMRs is observed within our lines, with no hypermethylated and only 15 hypomethylated DMRs shared amongst all patients (Fig. S2f).

We then compared gene expression levels derived from the RNA sequencing data of the cell lines generated in this study, two reference ESC lines[30] and one additional study of iPSC that also sequenced parental cell lines[31]. Hierarchical clustering of these datasets again demonstrated that our EPSC lines cluster closely with ESC (Fig. S2g). Comparative expression analysis revealed 250 differentially expressed (DE) genes between the two reference ESC lines. The numbers of DE genes between EPSC with ESC are more variable than with the methylation profiles, with increased numbers of upregulated DE genes in two of our EPSC lines and one of the reference iPSC lines (Fig. S2h). Integration of the DNA methylation and transcriptomic datasets revealed a low level of association between DMR and DE genes: only 11 genes appear in both datasets and none of those are shared across all patient-derived cell lines (Table S2). This remarkable epigenetic similarity of our EPSC lines to ESC was corroborated by their trilineage differentiation capacity (ectoderm, endoderm, and mesoderm) in an embryoid body formation assay (Fig. 1c, bottom). EPSC were then induced to differentiate into iNSC using a commercially available protocol (denoted iNSC^Gibco). Genome-wide expression

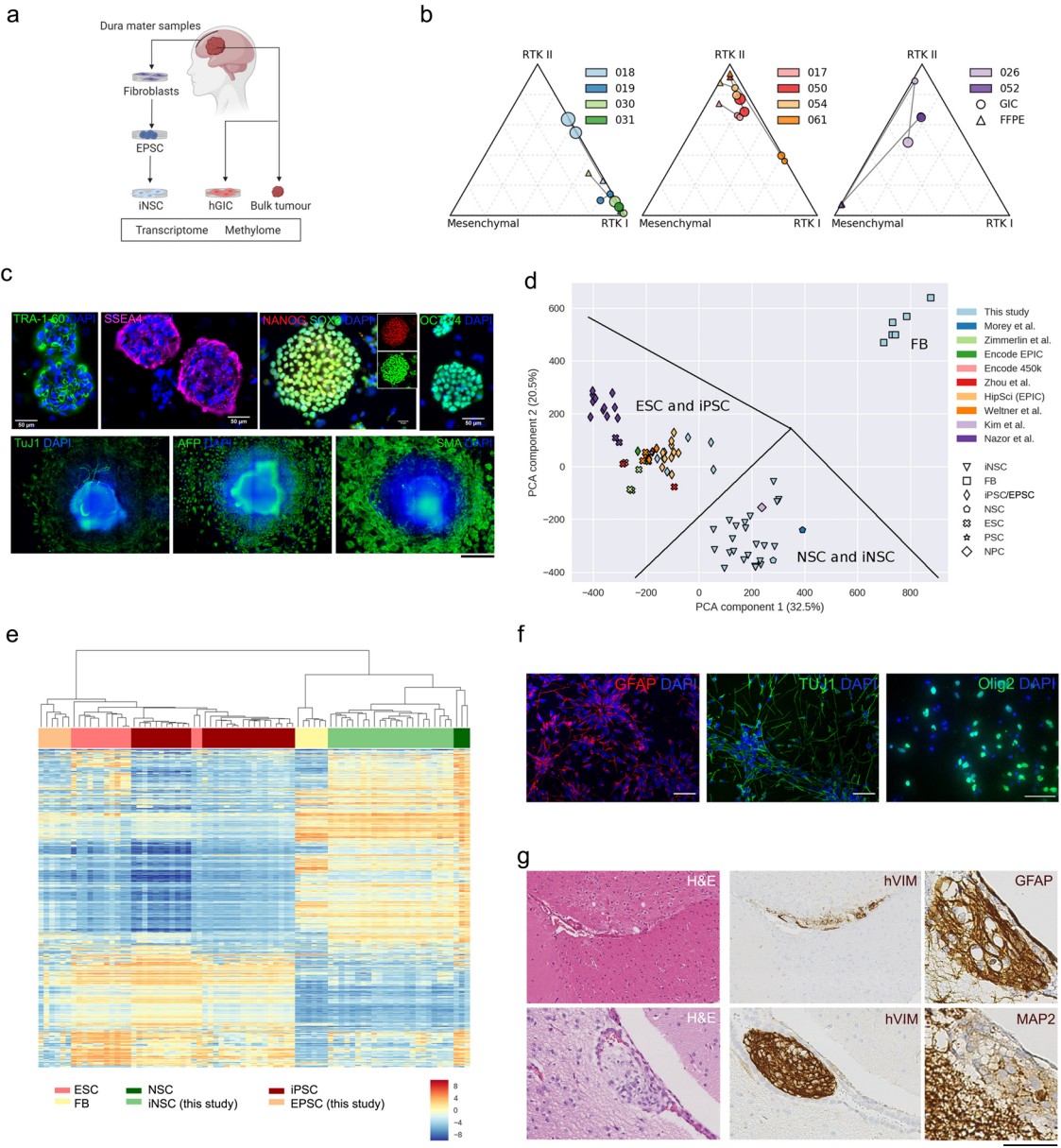

**Fig. 1 Generation and characterization of GIC and patient-matched EPSC-derived iNSC from human tissue samples. a** Schematic of the experimental setup. EPSC: expanded potential stem cells, iNSC: induced neural stem cells, GIC: glioblastoma initiating cells. **b** Subgrouping (receptor tyrosine kinase (RTK) I and II, mesenchymal) based on DNA methylation and a methylome classifier[18]. Triangles and circles represent GBM-FFPE and GIC respectively. Size corresponds to passage number; higher passages are represented as larger circles/triangles. Colours represent the different patients. **c** Expression of pluripotency cell surface markers (TRA1–60, SSEA4) and nuclear markers (SOX2, OCT3/4, and NANOG) in EPSC (top). EPSC-derived embryoid bodies expressing the three germ layer markers: neuron-specific class III beta-tubulin (TuJ1, ectoderm), alpha-fetoprotein (AFP, endoderm), and smooth muscle actin (SMA, mesoderm) (bottom). Experiments have been performed on each of the 10 EPSC lines with consistent results. The scale bar is 400 μm. **d** The first two principal components of the methylome of the EPSC, iNSC, and fibroblast lines included in this study together with reference datasets. Annotations have been added manually to guide interpretation. FB: fibroblasts, i/EPSC: induced pluripotent/expanded potential stem cells, ESC: embryonic stem cells, PSC: pluripotent stem cells, NPC: neural progenitor cells. **e** Hierarchical clustering of lines generated in this study and reference DNA methylation datasets. **f** Expression of astrocytic (GFAP), neuronal (TUJ1), and oligodendroglial (OLIG2) markers upon differentiation of iNSC. Experiments have been performed on 3 iNSC lines with consistent results. The scale bar is 50 μm. **g** Human VIMENTIN positive cells in the SVZ upon intracerebral injection of iNSC into adult NODSCID mice; expression of glial (GFAP) and early neuronal progenitor cells (MAP2). Experiments have been performed on 3 iNSC lines with consistent results. Scale bar is 250 μm for Haematoxylin and Eosin (H&E) (left) and hVIM (middle) and 125 μm for GFAP and MAP2 (right).

and methylation analysis confirmed that iNSC$^{Gibco}$ cluster together with various reference NSC lines and distinctly from iPSC and fibroblasts (Fig. 1d, e). These cells retained a normal karyotype (Fig. S2b), expressed NSC markers at protein level (Fig. S2i), formed neurospheres (Fig. S2j), downregulated the expression of pluripotency markers (Fig. S2k) and differentiated

into all brain lineages in vitro (Fig. 1f). Importantly they homed to the subventricular zone (SVZ) upon intracerebral injection in adult mice (Fig. 1g).

In summary, we show that GIC and syngeneic iNSC pairs can be established from surplus surgical tumour material and a strip of dura mater, respectively.

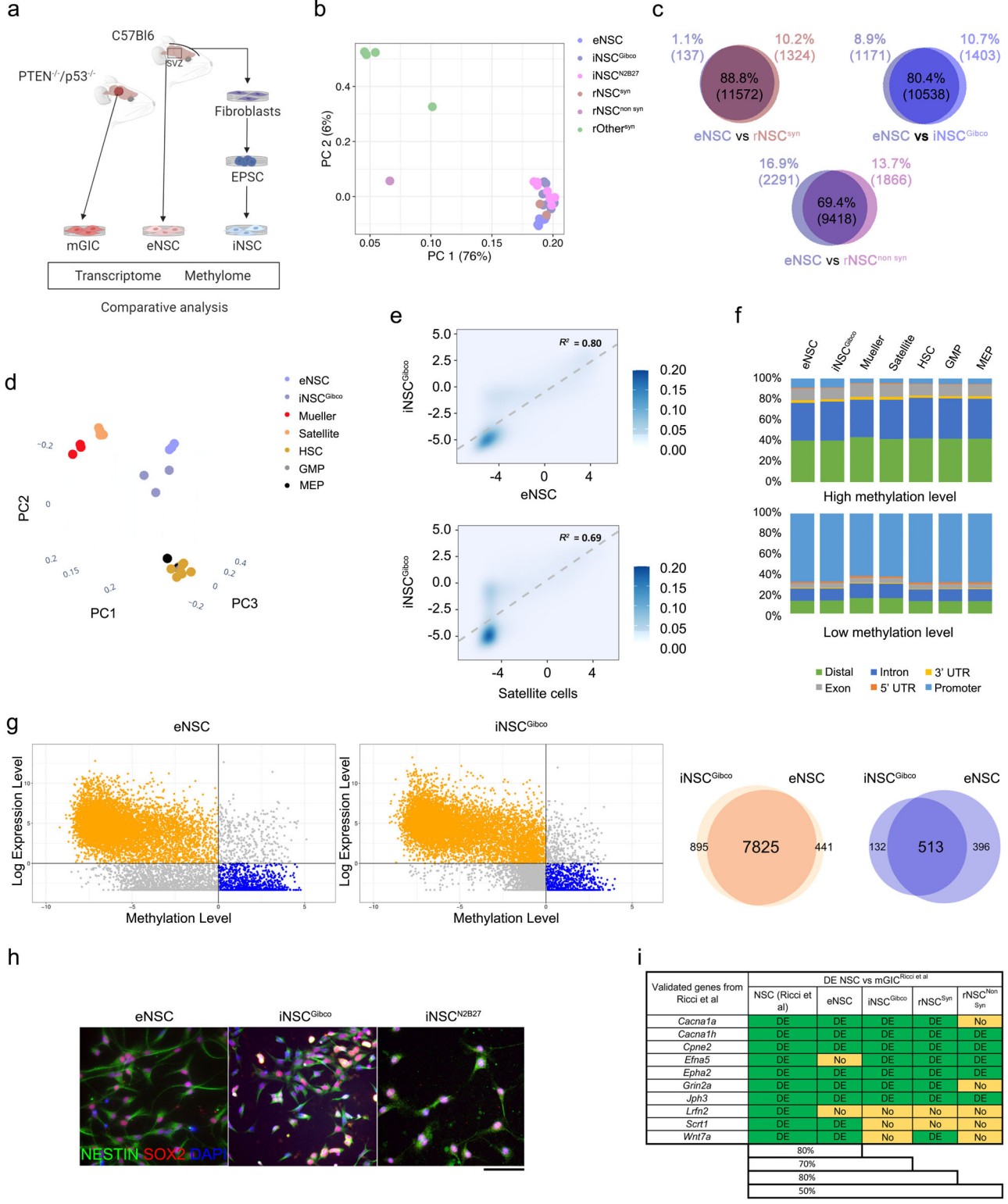

**iNSC are a suitable proxy for endogenous NSC to identify genes differentially expressed in a GBM mouse model**. To assess the biological relevance of iNSC as a surrogate for endogenous NSC (eNSC) to study GBM biology, we derived eNSC from the SVZ of three adult mice, as well as fibroblasts from the dura mater of the same mice (Fig. 2a and Fig. S3a). Fibroblasts were reprogrammed into EPSC, as described above, which was confirmed by the expression of stemness markers at RNA and protein levels (Fig. S3b, c) as well as their ability to form

embryoid bodies (EB) (Fig. S3d). These EPSC were then induced into NSC by the same protocol used in the human context above (denoted iNSC^Gibco) and also by a previously published customized protocol[32] which had been validated in the mouse context (denoted iNSC^N2B27). Comparative transcriptomic analysis clustered eNSC, iNSC^Gibco and iNSC^N2B27 together and close to reference NSC derived from mice of the same genetic background[33,34] (C57Bl6, denoted rNSC^syn) but separately from a reference NSC derived from non-syngeneic mice[35] (129sv/ev,

**Fig. 2 EPSC-derived iNSC are a suitable proxy for eNSC in comparative studies with GIC in mice. a** Schematic of experimental workflow. EPSC: expanded potential stem cells, iNSC: induced neural stem cells, eNSC: endogenous neural stem cells, mGIC: mouse glioblastoma initiating cells. **b** Two dimensional principal component analysis of cell types, representing normalized RNA sequencing data of eNSC, iNSC$^{Gibco}$, iNSC$^{N2B27}$, two rNSC$^{syn}$, one rNSC$^{non\ syn}$, two astrocytes, one neuron, and one GMP (granulocyte/monocyte progenitor). Astrocytes, neuron, and GMP are labelled as rOther$^{syn}$. e/i/r NSC: (endogenous/induced/reference neural stem cells, Gibco: EPSC neural induction with Gibco commercial protocol. N2B27: EPSC neural induction with a bespoke published protocol[32], syn: syngeneic, non-syn: non-syngeneic. **c** Venn diagrams of overlapping genes (TPM > 1) in iNSC$^{Gibco}$ and eNSC, rNSC$^{syn}$, rNSC$^{non\ syn}$ or neurons; results are expressed as percentages of genes shared or specific to each type of cells. **d** Three-dimensional principal component analysis of cell types, representing genome-wide methylation RRBS-Seq data ($M$-values) of eNSC, iNSC$^{Gibco}$, Mueller cells, muscle satellite cells, hematopoietic stem cells (HSC), granulocyte-macrophage progenitor (GMP), and megakaryocyte/erythrocyte progenitors (MEP). **e** Two-dimensional density plots of DNA methylation profiles ($M$-values) between iNSC$^{Gibco}$ and eNSC (top) and between iNSC$^{Gibco}$ and muscle satellite cells (bottom). A median distribution of $M$-values has been obtained for each cell type across replicas before performing Spearman's correlation. The $R^2$ value is shown for each plot. **f** Bar plots showing the relative abundance of genomic regions (distal, exon, intron, promoter, 3′ UTR, 5′ UTR) in methylated (top) and unmethylated (bottom) CpG sites across all cell types. **g** Left: combined analysis of expression profile and methylation levels of eNSC and iNSC$^{Gibco}$ (11107 genes are plotted). Genes are considered expressed if log (TPM) > 0 and methylated if $M > 0$. Concordant genes are depicted in orange and blue, according to the orientation. Right: Venn diagrams showing the overlap of concordant genes between eNSC and iNSC$^{Gibco}$ for both orientations. **h** Coexpression of SOX2 (red) and NESTIN (green) in eNSC, iNSC$^{Gibco}$, and iNSC$^{N2B27}$ with DAPI nuclear counterstaining (representative images, $n = 3$ for each cell type). The scale bar is 20 μm. **i** Percentage of genes classified as Differentially Expressed (DE) and not DE in comparisons between GIC vs NSC[39] or mGIC vs eNSC, iNSC$^{Gibco}$, rNSC$^{syn}$, and rNSC$^{non\ syn}$ respectively.

denoted rNSC$^{non\ syn}$) as well as other brain (astrocytes and neurons) and non-brain (monocytes) cells derived from syngeneic mice[36–38] (C57Bl6, denoted rOther$^{syn}$) (Fig. 2b and Fig. S4). In particular, the driver of clusters, the principal component PC1, clearly separates eNSC, iNSC, and rNSC$^{non\ syn}$ from the other cell types and is not able to discriminate eNSC from iNSC and the syngeneic reference (Fig. 2b). In agreement with these results, the highest overlap of expressed genes was noted between eNSC and rNSC$^{syn}$ (88.8%), followed by iNSC$^{Gibco}$ and iNSC$^{N2B27}$ (85.4%) and eNSC and iNSC$^{Gibco}$ (80.5%) (Fig. 2c and Fig. S3e). The degree of overlap reduced to 69.4% when iNSC$^{Gibco}$ were compared with non-syngeneic reference NSC (Fig. 2c), and to 64.7 and 61.5% when eNSC and iNSC$^{Gibco}$ were compared to syngeneic neurons, respectively (Fig. S3e).

Comparative analysis of global DNA methylation profiles between eNSC, iNSC$^{Gibco}$ and syngeneic non-brain cells shows clustering of eNSC and iNSC$^{Gibco}$ (Fig. 2d) and higher correlation between iNSC$^{Gibco}$ and eNSC than the other cell types, including stem cells of mesenchymal origin (Fig. 2e and Fig. S3f). We compared the proportions of the annotated regions between cell types and showed that the probability of those proportions being different is lowest between eNSC and iNSC$^{Gibco}$, highlighting a higher degree of similarity compared to the other cell types (Fig. 2f and Fig. S3g). Additionally, comparative analysis of combined methylome and transcriptome of eNSC and iNSC$^{Gibco}$ confirmed a high degree of overlap (82%) of concordant genes (methylated and low expression and vice-versa) (Fig. 2g).

Phenotypic and functional characterization of eNSC, iNSC$^{Gibco}$, and iNSC$^{N2B27}$ confirmed coexpression of the NSC markers Nestin and Sox2 (Fig. 2h) and their ability to form neurospheres (Fig. S3h).

Finally, we queried the value of these iNSC in identifying DE genes in a glioblastoma mouse model[39]. We show that syngeneic iNSC$^{Gibco}$ identified more DE genes as compared to non-syngeneic NSC (rNSC$^{non\ syn}$), 70% as compared 50% (Fig. 2i).

Our results in mouse models show that iNSC are reasonably similar to eNSC isolated from the SVZ and that syngeneic iNSC are superior to non-syngeneic iNSC in identifying deregulated genes in GBM.

**Comparative transcriptome analysis of syngeneic GIC/iNSC pairs identifies a GIC-specific GAG-mediated regulatory T cells recruitment mechanism in a proportion of patients.** Comparative analysis of syngeneic GIC/iNSC pairs identified genes

differentially expressed in individual patients; these were then analyzed for molecular pathway enrichment on the Ingenuity Pathway Analysis platform (Fig. S5a). Only one pathway was enriched for in all patients (Hepatic fibrosis, which comprises *PDGF/TNF/VEGFR/EGFR/FGF*, all molecular cascades known to be deregulated in GBM), three pathways were shared among the majority of patients (Axonal Guidance Signalling, Antigen Presentation Pathway and Glutamate Receptor Signalling) but the majority of pathways were enriched for in individual or few patients (Fig. S5a). Importantly, significantly more pathways were identified when using syngeneic iNSC as a comparator rather than non-syngeneic commercially available iNSC lines (Gibco and H9, Fig. S5b). Moreover, several pathways were specifically enriched for when the syngeneic comparator was used but not with non-syngeneic iNSC, such as heparan sulfate (HS) biosynthesis (GIC19, 31, 50), chondroitin sulfate (CS), and dermatan sulfate (DS) biosynthesis (GIC19, 31, 50), Opioid signalling pathway (GIC19, 50 and 54), GABA receptor signalling (GIC18), Putrescin Degradation (GIC19 and GIC26), Paxillin Signalling (GIC54, 52) (Fig. 3a and Figs. S5a, S6a). Because network visualization highlighted the deregulation of pathways involved in cell−cell interactions, tumour inflammation, and modulation of the microenvironment in a subset of patients (Fig. 3a), we set out to study the cellular composition of the GBM bulk tumour using xCell, an in silico analytical tool based on transcriptomic data[40]. We determined the correlation between the inferred cell type proportion of the bulk tumour (Fig. S6b) and the pathway enrichment in the genes DE in hGIC relative to iNSC (Fig. S6a, c). A significant correlation was noted between enrichment for pathways linked to the biosynthesis of GAGs, linear polysaccharides including HS and CS/DS, in the hGIC/iNSC comparison, and higher proportion of regulatory T-cells (Tregs) as well as a lower proportion of macrophages in the corresponding bulk tumour (Fig. S6c and Fig. 3b). This correlation was not identified when a non-syngeneic iNSC comparator was used for the identification of hGIC DE genes (Fig. S6d). Immunostaining for the Tregs marker FoxP3 and CD4 in the corresponding bulk tissue samples confirmed a higher number of Tregs among the CD4+ tumour infiltrating lymphocytes (TIL) (Fig. 3c), which was most prominent in those patients predicted to have a higher proportion of Tregs as assessed by CS/DS biosynthetic pathway deregulation (Fig. S7a). Increased levels of CS in GIC as compared to the matched iNSC were confirmed using immunocytochemistry and were neutralized by targeted digestion using chondroitinase ABC enzyme (chABC) (Fig. S7b). We

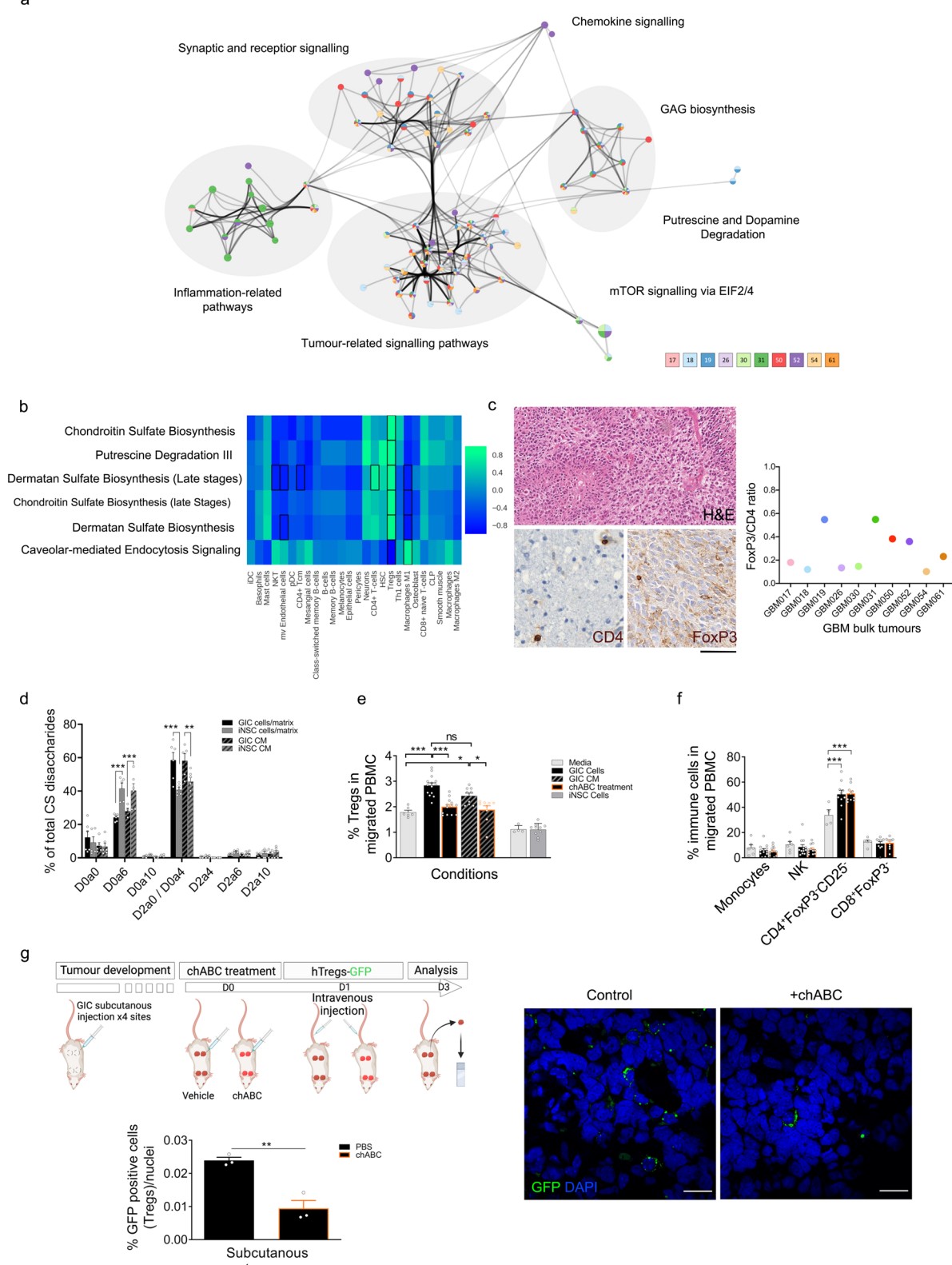

hypothesized that deregulated GAG biosynthesis may influence Tregs infiltration in a proportion of GBM patients, which contributes to the generation of an immunosuppressed microenvironment in GBM[41]. Biosynthesis of CS/DS involves the activity of multiple sulfotransferases and other synthetic enzymes. To read out this activity, we analyzed the composition of GAGs synthesized by iNSC and GIC cultures. GAGs were isolated from

GIC and iNSC cell extracts and conditioned media, digested to disaccharides, and analyzed by RP-HPLC (Fig. 3d and Fig. S7c). We observed an increase in 6-O-sulphation (D0a6) in iNSC compared to GIC, with a correspondingly lower proportion of HexA(2S)-GalNAc (D2a0) and HexA-GalNAc(4S) (D0a4) disaccharides (Fig. 3d). A similar pattern was observed in both cell/matrix derived CS/DS and CS/DS isolated from the conditioned

**Fig. 3 Comparative transcriptome analysis of GIC/iNSC identifies a GAG-mediated mechanism of Tregs migration in a proportion of GBM. a** Network representation of the enriched pathways identified in the 10 syngeneic GIC-iNSC comparisons. Nodes represent pathways; pie charts are shaded according to the patients in which the pathway is identified (see legend); size is proportional to the mean value of −log10(FDR) across all patients. Edges represent deregulated genes shared between pathways (pooled over relevant patients); thickness is proportional to the number of genes shared. **b** Heatmap showing the Spearman rank correlation across patients between pathway enrichment in the GIC culture relative to matched iNSC, expressed as –log10($p$), and estimated cell type composition in the corresponding GBM bulk tissue sample. Cells with statistically significant correlation coefficient are highlighted ($p < 0.05$). **c** Representative histology of glioblastoma bulk tissue (Hematoxylin & Eosin (H&E) staining, top), immunostaining for CD4 (bottom left) and FoxP3 (bottom right). FoxP3/CD4 ratio in all GBM samples (right). Each case has been stained once. Scale bar is 250 μm. **d** Disaccharide composition of CS isolated from GIC (black bars) and iNSC (grey bars) cell extracts (plain bars) and from conditioned media (hatched bars) analyzed by RP-HPLC. D0a0: HexA-GalNAc; D0a6: HexA-GalNAc(6S); D2a0 HexA(2S)-GalNAc; D0a4: HexA-GalNAc(4S); D2a6 HexA(2S)-GalNAc(6S); D0a10: HexA-GalNAc(4S, 6S). Results are an average of two GIC/iNSC pairs, 19 and 31, ($n = 2$, two way ANOVA). (D0a6: $p$ value (GIC vs iNSC cells/matrix) < 0,0001; $p$ value (GIC vs iNSC CM) = 0,0009; D2a0/D0a4: $p$ value (GIC vs iNSC cells/matrix) <0,0001, $p$ value (GIC vs iNSC CM) = 0,0003). Percentages of migrated immune cells in a transwell assay in the presence of GIC (black bars), their 24 h conditioned-media (CM, hatched black bars), iNSC (grey bars) and media only (light grey bars) after 4 h incubation. Percentage of Tregs (CD4+CD8-FOXP3+CD25+CD127−, **e** and other immune cells (CD14+ monocytes, CD56+natural killer cells, CD4+Foxp3−Cd25− and CD8+Foxp3− cells, **f** with (orange border) and without chondroitinase ABC (chABC) pre-treatment. Results are obtained from patients 19 and 31, each experiment was repeated 4−7 times (GIC media: $n = 6$, NSC media: $n = 4$, GIC: $n = 7$, GIC CM: $n = 5$, GIC + Chase: $n = 7$, GIC CM + Chase: $n = 5$, iNSC: $n = 5$, one-way ANOVA). **e** $p$ value (Media vs GIC cells) < 0,0001; $p$ value (Media vs GIC CM) = 0,0103; $p$ value (GIC cells vs chABC) < 0,0001; $p$ value (GIC CM vs chABC CM) = 0,0184; $p$ value (GIC cells vs GIC CM) = 0,0806. **f** CD4+Foxp3−CD25+: $p$ value (Media vs GIC) = 0,0004, $p$ value (Media vs GIC chABC) = 0,0002. **g** Schematic of experimental workflow (top left), percentage of GFP-labelled hTregs over total cell number per tumour section upon treatment with vehicle (control) or chondroitinase ABC (chABC) (bottom left), ($n = 4$, two tailed t-test $p$ value = 0,0049). Representative GFP immunofluorescence, scale bar is 20 μm. All graphs report mean ± SEM. Statistical significance for all panels *$p \leq 0.05$, **$p \leq 0.01$, ***$p \leq 0.001$. Source data are provided in the source data file.

media. This was not the case for HS, a GAG that shares only early elements of the CS/DS biosynthetic pathway. The distribution of HS disaccharides was similar between cell/matrix and media samples in iNSC, however, these were markedly different in GIC (Fig. S7c). HS from GIC conditioned media had increased levels of 2-O-sulfation and 6-O-sulfation (D2S0 and D2S6) compared to iNSC conditioned media.

Next, we set out to assess the functional impact of the observed shift in GAG sulfation on Tregs by means of in vitro chemotaxis experiments. We showed increased migration of Tregs in the presence of GIC (black bars) compared to their matched-iNSC (dark grey bars) (Fig. 3e and Fig. S7d), an effect which was predominantly mediated by the cellular fraction as it was reduced when PBMC were incubated with GIC-conditioned media alone (hatched bars). This effect of GIC on Tregs migration was dependent on CS/DS as pretreatment with chABC, catalyzing digestion of CS and DS chains (Fig. 3e), blocked Tregs migration in the transwell assay (black bar with orange border). The observed GAG dependent Tregs migration is specific to this cell population and does not impact CD4+Foxp3−CD25− T cells (Fig. 3f and Fig. S7e), CD8+Foxp3− (Fig. 3f and Fig. S7f) and monocytes (Fig. 3f and Fig. S7g) or NK cells (Fig. 3f and Fig. S7h). iNSC did not affect the migration of all tested immune cells (Fig. S7i). Depletion of CS/DS in subcutaneous GIC19-derived xenografts led to decreased Tregs migration to the tumour site as compared to vehicle-treated tumours (Fig. 3g and Fig. S7j).

In summary, differential transcriptional analysis of GIC/iNSC syngeneic pairs identified a shift in GAG sulfation in a proportion of patients, which controls Tregs migration in GBM.

**Integrated transcriptome and methylome analysis of syngeneic hGIC/iNSC pairs identifies patient-specific druggable targets.** To explore the potential suitability of the platform to identify genes for which compounds exist, and which could be tested for patient selective efficacy, we set out to comparatively interrogate transcriptome and methylome profiles (Fig. 4a). The process yielded around 17,000 unique (gene, methylation cluster) pairs corresponding to around 12,000 unique genes (the background). We identified 2,222 unique differentially expressed pairs with a corresponding differentially methylated region in at least one patient, of which 1,565 (70%) were concordant, corresponding to

1,274 unique genes (Supplementary Data 1). Of the concordant pairs, 733 (47%) were specific to one patient corresponding to 639 distinct genes, ranging from 20 pairs in patient 50−177 in patient 18 (Fig. S8a). In contrast, a very small proportion (17 pairs, 1%) was shared across all patients. Furthermore, no single combination of patients shared a substantial number of DE/DMR pairs: all subsets had fewer than 25 (Fig. S8a). This emphasizes the high inter-patient heterogeneity of glioblastoma and the importance of a patient-specific approach.

We queried the drug-gene interaction database (DGIdb)[42] against the 1,274 gene list and identified 1,920 distinct drug compounds known to interact with 237 of them (19%) (Supplementary Data 1). This is a highly significant enrichment relative to the <3% of background genes with known interactions (Fisher's test $p < 10^{-6}$), confirming that our analytical pipeline is an effective approach to enrich for druggable targets. 645 (34%) of the compounds are predicted in a single patient, ranging from 15 in patient 17 to 205 in patient 18, further highlighting the importance of a patient-specific approach (Fig. S8b). This corresponds to 639 unique genes, of which 118 (18%) have matching drug compounds and are therefore candidates for potential follow-up (Supplementary Data 1).

In order to further prioritize targets, we shortlisted a subset with the even more stringent criterion that genes must be patient-specific at the level of DE and DMR independently (Fig. 4a). While genes in the longlist may be DE or DMR in multiple patients providing that only one has the combination of both DE and DMR, genes in the shortlist are patient-specific independently at the level of DE and DMR. This approach yields 53 unique genes, of which 15 (28%) have matching drug compounds. Taking into account the mechanism of action (an agonist for a gene hypermethylated/downregulated and an antagonist for a gene hypomethylated/overexpressed), commercial availability of the compound, and their pharmacological properties (specificity, potential for blood barrier crossing), PGE1-OH targeting PTGER4 (patient 18) and Disulfiram (DSF) targeting ALDH3B1 (patient 19) were selected for experimental validation. NTRK2, identified in patient 30, was also taken forward to validation as it was predicted to be effectively targetable with Cyclotraxin-B (CTX-B). Interrogation of a publicly available collection of 71 GICs with DNA methylome

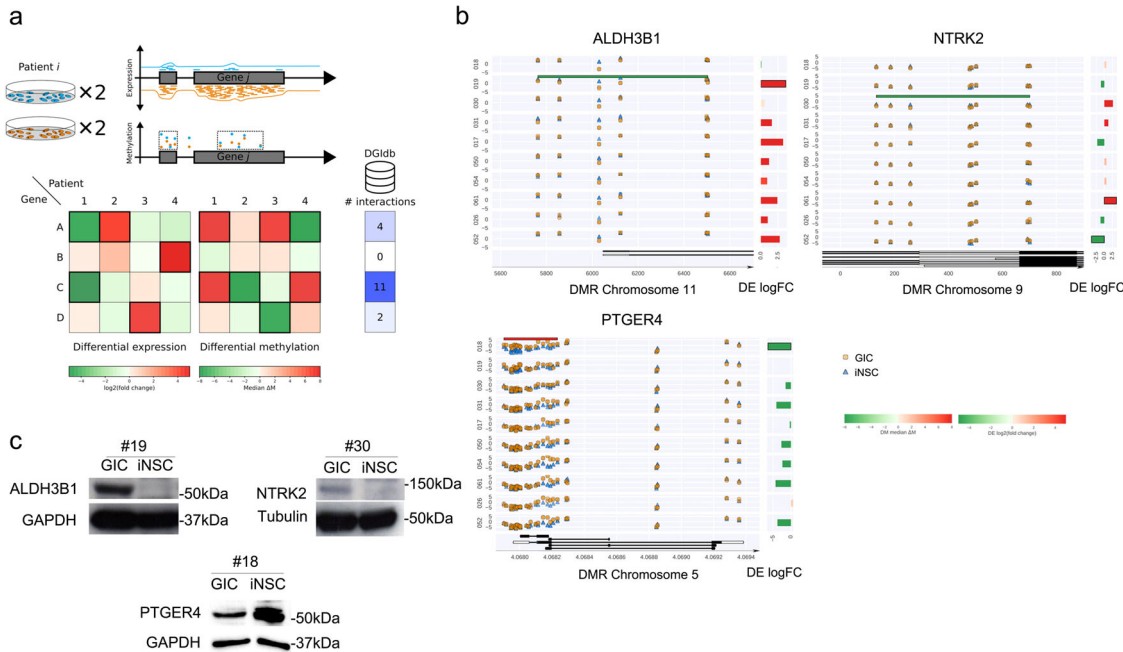

**Fig. 4 Comparative analysis of transcriptome and methylome of GIC/iNSC identifies patient-specific drug sensitivities in silico. a** Schematic of analysis conducted to combine gene expression from RNAseq and DNA methylation data from EPIC array together with a screen for drug compounds. DGIdb: drug gene interaction database. **b** Differentially methylated region (DMR, from DNA methylation sequencing) between GIC (orange circles) and iNSC (blue triangles) and on the right the differentially expressed genes (−logFC from RNA sequencing data). Plots represented show selected genes with concordance between methylation and expression data for one patient only: *ALDH3B1* specific of patient 19 (top left), *NTRK2*, specific of patient 30 (top right), and *PTGER4*, specific of patient 18 (bottom left). DE differential expression, DMR differential methylated region, log FC log fold change. **c** Protein expression for ALDH3B1, NTRK2, and PTGER4 in GIC and iNSC of their respective pair of interest (19, 30, and 18). Tubulin and GAPDH were used as housekeeping proteins for NTRK2 and ALDH3B1/PTGER4 respectively. Experiments have been performed three independent times with consistent results.

and transcriptome datasets (HGCC) identified 34% of cultures displaying high methylation/low expression of PTGER4 as compared to cohort average, as well as 21 and 22.5% low expression/high methylation of NTRK2 and ALDH3B1 respectively, raising the possibility that these signatures could be widely shared among GBM patients (Fig. S8c).

We show that druggable target genes can be identified in a proportion of patients by integrative analysis of the DNA methylome and transcriptome of GIC as compared to their syngeneic iNSC.

**Patient-specific efficacy of Prostaglandin E1 alcohol in 2D and 3D syngeneic GLICO models is predicted by methylation-mediated regulation of PTGER4 expression.** *PTGER4* is the gene encoding the receptor for prostaglandin E2 (PGE2) and it belongs to the G-protein coupled receptor family. It is hypermethylated with a concordant, strongly significant downregulation in GIC18 as compared to the syngeneic iNSC (Fig. 4b), and reduced expression of the gene was confirmed at protein level (Fig. 4c). It has been suggested that it serves as a proliferation inhibitor uniquely in GIC that maintains a pool of slow-cycling cells refractory to conventional therapy[43]. In agreement with this hypothesis, higher expression of *PTGER4* is observed in the tumour bulk of GBM18 and in the TCGA GBM cohort (Fig. S9a) as compared to GIC18, and a negative correlation was observed between the expression of *PTGER4* and *PROM1* (*CD133*), a GIC marker, in the TCGA datasets (Fig. S9b). Moreover, genetic overexpression of *PTGER4* in GIC18 (Fig. S9c) negatively impacted the proliferation of the cells (Fig. S9d), in keeping with the interpretation that it has a tumour suppressive role in GIC. Prostaglandin E1 alcohol (PGE1-OH) is a specific EP4 agonist used clinically as a non-irritant bronchodilator[44,45]. To assess

whether the in silico prediction of its efficacy in patient 18 could be validated in vitro, we used patient 18 (low expression and hypermethylation in GIC as compared to iNSC, where only the GIC are predicted to respond to PGE1-OH), patient 30 (low expression of PTGER4 in both GIC and iNSC and no concordant differential methylation, both predicted to respond to PGE1-OH) and patient 26 (high expression of PTGER4 in both GIC and iNSC and no concordant differential methylation, both predicted not to respond to PGE1-OH) (Fig. S9e and Table S3). We show that treatment with PGE1-OH significantly reduces viability in GIC18 without affecting the matched iNSC (Fig. 5a, top), while it had a similar effect on GIC30 and iNSC30 (Fig. 5a, bottom). For patient 26, both GIC and iNSC did not respond to treatment with PGE1-OH (Fig. S9f), in keeping with the in silico prediction.

PGE1-OH treatment decreases proliferation specifically in GIC18 (Fig. 5b), without triggering toxicity (Fig. S9g) or apoptosis (Fig. S9h) and without affecting tumour sphere formation (Fig. S9i).

Next, we used a catalytically inactive Cas9 (dCas9) fused with Tet1 (ten-eleven translocation Methylcytosine Dioxygenase 1) to assess the functional relevance of *PTGER4* hypermethylation by site-specific DNA methylation editing[46]. We show successful demethylation of an hypermethylated CpG dinucleotide within a CpG island in the *PTGER4* promoter (cg04727116, Illumina Infinium Methylation EPIC) in GIC18 (GIC18$^{TET1/sgRNA4}$) (Fig. 5c and Fig. S10a) which resulted in increased expression of *PTGER4* (Fig. S10b). GIC18$^{TET1/sgRNA4}$ showed reduced proliferation compared to non-edited cells (Fig. 5d, black bars vs hatched black bars) and mice bearing a GIC18$^{TET1/sgRNA4}$ derived xenograft survived longer as compared to mice xenografted with GIC18$^{TET1/pgRNA}$ (Fig. 5e). Importantly, no sensitivity to PGE1-OH treatment was seen in GIC18$^{TET1/sgRNA4}$ (Fig. 5d grey vs hatched grey bars).

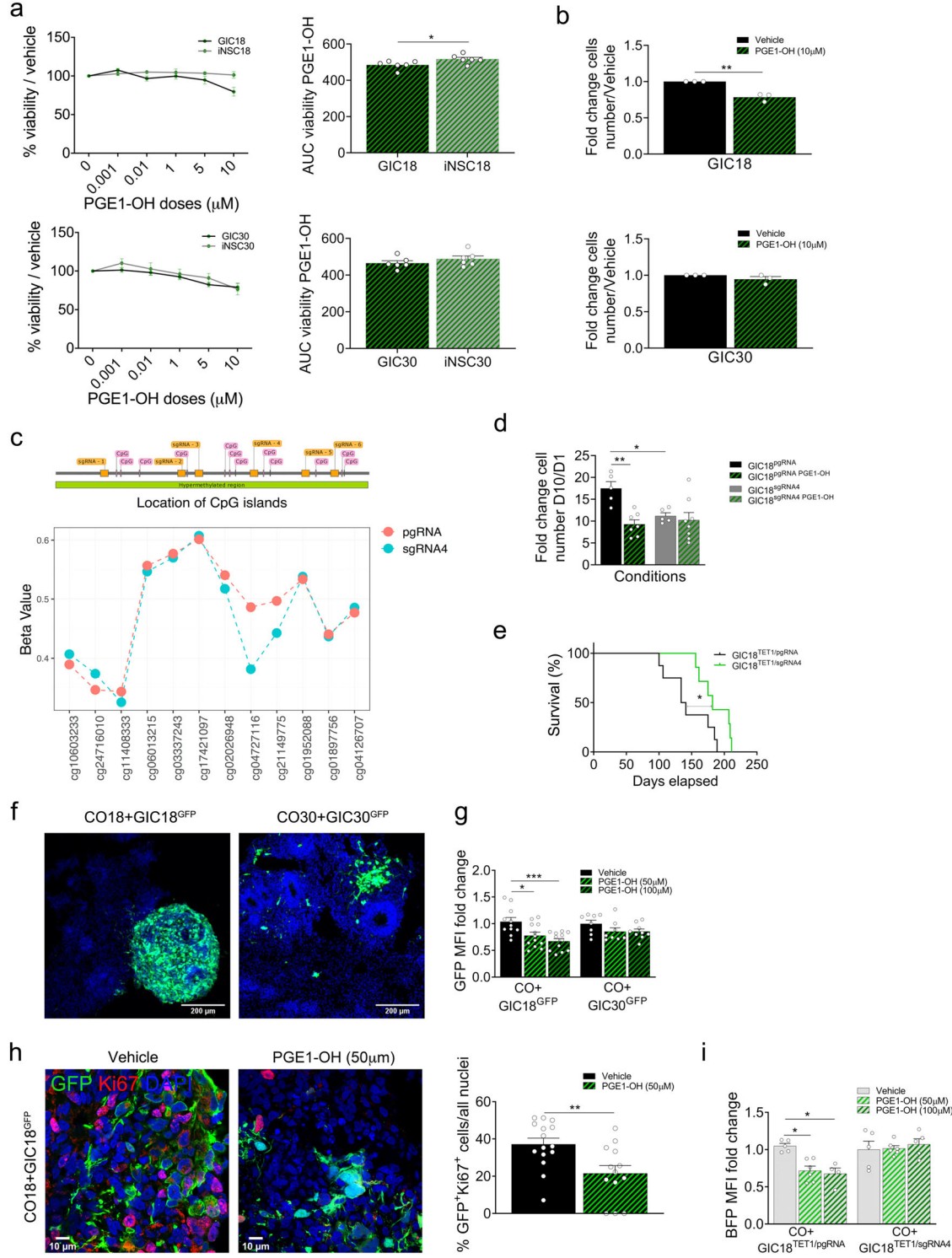

PGE1-OH is administered to patients via inhalation[45], but we are not aware of any other use of either PGE1-OH or other PTGER4 agonists, hence making in vivo validation not viable. Therefore, to provide further support to the selective efficacy of the drug in patient 18, we used the 3D cerebral organoid glioma (GLICO) model whereby GIC are grown within a cerebral organoid (CO), which can be used as an effective predictor of drug response in vivo[47,48]. Taking advantage of the availability of patient-matched EPSC we set up a modified GLICO model where GFP-labelled GIC are co-cultured with CO derived from the matched EPSC (SYNGLICO, Fig. 5f). CO developed successfully,

as demonstrated by multiple neural rosettes observed throughout the CO on morphology, with neuroectodermal cells, expressing SOX2 and NESTIN, arranged in radial fashion within the rosettes and NeuN-positive cells populating the neuropil between the neural rosettes (Fig. S10c). To assess the response to PGE1-OH, 100,000 GIC18^GFP and GIC30^GFP were incubated with 30 days old EPSC18-derived and EPSC30-derived CO respectively, for 5–7 days and treated with 50 and 100 μM PGE1-OH for 4 days. We show no significant difference in the overall viability of the co-cultures (Fig. S10d, e), while significantly decreased mean fluorescent intensity (MFI) of GIC18^GFP but not GIC30^GFP was

**Fig. 5 Comparative analysis of transcriptome and methylome of GIC/iNSC identifies PTGER4/PGE1-OH as patient-specific drug sensitivity. a** Drug treatment of GIC (black curves) and iNSC (grey curves) from patient 18 (top) and 30 (bottom) with increasing doses (1 nM−10 μM) of PGE1-OH (green dots) as percentage of viability on the vehicle, measured at end point; area under the curve (AUC) was calculated from the percentages of viability ($n = 6$ experiment repetitions, two tailed t-test, $p$ value (patient 18) = 0,0434, $p$ value (patient 30) = 0,3037). **b** Proliferation assay of GIC18 (top) and GIC30 (bottom) expressed as cell numbers after 4 days of treatment with PGE1-OH (green hatched bars) on cell number at day 1, standardized on the vehicle (black bars). ($n = 3$ experiment repetitions, two-tailed t-test; $p$ value (patient 18) = 0,0026, $p$ value (patient 30) = 0,2172). **c** Map of the CpG island and guides RNA (top) and differentially methylated regions (DMR) between GIC18$^{TET1/pgRNA}$ (red dots) and GIC18$^{TET1/sgRNA4}$ (blue dots) (bottom). Y-axis reports Beta values and location on Infinium MethylationEPIC array is on x-axis. **d** Proliferation assay of GIC18 transduced with control dCas9-*TET1* + pgRNA (black bars) or with dCas9-*TET1* + sgRNA4 (grey bars) expressed as fold change of cell numbers after 10 days of treatment with PGE1-OH (10 μM) (green hatched bars) on day 1 and standardized on the vehicle treatment. ($n = 3$ experiment repetitions, two-tailed t-test; pgRNA: $p$ value (Veh vs PGE1-OH) = 0,0039, $p$ value (pgRNA vs sgRNA4) = 0,0485). **e** Survival curve of mice bearing GIC18$^{TET1/pgRNA}$ and GIC18$^{TET1/sgRNA4}$-derived intracerebral xenograft (GIC18$^{TET1/pgRNA}$: $n = 8$, GIC18$^{TET1/sgRNA4}$: $n = 7$, log-rank (Mantel−Cox) test, two-sided; $p$ value = 0,0426). **f** Representative GFP immunofluorescence of SYNGLICO18 and 30 (GIC$^{GFP}$ cultured with their syngeneic CO (cerebral organoid)). The scale bar is 200 μm. **g** GFP mean fluorescence intensity (MFI) of GIC$^{GFP+}$ cells (SYNGLICO18 vehicle: $n = 11$, PGE1-OH 10 and 50 μM: $n = 12$; SYNGLICO30 vehicle and PGE1-OH 50 μM: $n = 8$, PGE1-OH 100 μM: $n = 9$ per group, one-way ANOVA, SYNGLICO18: $p$ value (Vehicle vs PGE1-OH 50 μM) = 0,0357, $p$ value (Vehicle vs PGE1-OH 100 μM) = 0,0008). **h** Representative GFP and Ki67 immunofluorescence of SYNGLICO after 4 days of treatment with vehicle or PGE1-OH (50 μM). The scale bar is 10 μm. Quantification of GFP and Ki67 double-positive cells. ($n = 3$ SYNGLICO Vehicle: $n = 15$ fields, PGE1-OH 50 μM: $n = 13$ fields, two tailed t-test, $p$ value 0,0057). **i** BFP MFI of GIC18$^{TET1/pgRNA}$ and GIC18$^{TET1/sgRNA4}$ after 4 days of treatment with vehicle or PGE1-OH (50 μM, 100 μM) (SYNGLICO18$^{TET1/pgRNA}$ Vehicle: $n = 5$, PGE1-OH 50 μM, $n = 6$ and PGE1-OH 100 μM: $n = 4$; SYNGLICO18$^{TET1/sgRNA4}$ vehicle: $n = 5$ and PGE1-OH 100 μM $n = 5$, PGE1-OH 50 μM $n = 6$, one way ANOVA; pgRNA: $p$ value (Vehicle vs PGE-OH 50 μM) = 0,0195, $p$ value (Vehicle vs PGE-OH 500 μM) = 0,0174). All graphs report mean ± SEM. Statistical significance for all panels *$p \leq 0.05$, **$p \leq 0.01$, ***$p \leq 0.001$. Source data are provided in the source data file.

observed (Fig. 5g and Fig. S10e). Assessment of proliferation (KI67 immunostaining) and apoptosis (cleaved Caspase 3 immunostaining) in CO + GIC18$^{GFP}$ treated with 50 μM PGE1-OH confirmed reduced proliferation and increased apoptosis in GIC18$^{GFP}$ (Fig. 5h and S11a).

Finally, we confirmed in our SYNGLICO model that PGE1-OH effect was dependent on *PTGER4* methylation as PGE1-OH treatment had no effect on CO + GIC18$^{TET1/sgRNA4}$ (Fig. 5i and Fig. S11b, c).

Our results show that PTGER4 functions as a tumour suppressor in selected GBM patients and regulation of its expression by DNA methylation predict response to PTGER4 activation therapy in 2D and 3D in vitro models.

**Patient-specific efficacy of disulfiram and cyclotraxin-B in vitro and in vivo.** Aldehyde dehydrogenase 3 Family Member B1 (ALDH3B1), a member of the aldehyde dehydrogenases family protecting against oxidative stress[49], is hypomethylated specifically in GIC19 as compared to the syngeneic iNSC19 within our cohort (Fig. 4b). A concomitant upregulation of expression (Fig. 4b) was observed, which was confirmed at protein level (Fig. 4c). DSF, a specific inhibitor of the ALDH superfamily[50], is known to reduce cell viability, proliferation, and self-renewal capacity, as well as to induce cell cycle arrest and apoptosis in paediatric brain tumours[51].

We show in 2D monolayer cultures that DSF reduces cell viability (Fig. 6a, top) and induces cytotoxicity (Fig. S12a, top) in GIC19 without eliciting an effect on the syngeneic iNSC. Importantly, GIC31 and iNSC31 which did not show differential expression or methylation for *ALDH3B1*, showed no sensitivity to the drug (Fig. 6a, bottom, and Fig. S12a, bottom). To further dissect the impact of DSF treatment on cellular functions, we assessed proliferation, tumour sphere formation as well as apoptosis and showed decreased proliferation (Fig. 6b) and tumour spheres formation (Fig. 6c and Fig. S12c) whilst apoptosis was increased (Fig. S12b) in GIC19 but not in GIC31. Concomitantly, total ALDH activity was significantly more impacted by DSF treatment in GIC19 as compared to GIC31 (Fig. S12d), suggesting the effect of DSF was specifically on ALDH in our experimental setting. Silencing of *ALDH3B1* (Fig. S12e), which did not affect the expression of other *ALDH*

isoforms (Fig. S12f), inhibited GIC19 proliferation (Fig. S12g) and confirmed that the impaired viability observed after DSF treatment is ALDH3B1-dependent as it is lost in GIC upon ALDH3B1 silencing (Fig. S12h).

These results were confirmed in the 3D SYNGLICO model where DSF treatment impacted the MFI of GIC19$^{GFP+}$ cells within CO + GIC19$^{GFP+}$ without affecting viability (Fig. 6d, e and Fig. S13, left) indicating loss of GFP$^{+}$ tumour cells whereas no such effect was observed in GIC18$^{GFP+}$ CO (Fig. 6d, e and Fig. S13, right). Reduced proliferation and increased apoptosis were confirmed in GIC19$^{GFP}$ cultured within CO19 treated with 10 and 50 μM DSF by immunostaining for Ki67 and cleaved-caspase-3 respectively (Fig. 6f, g).

In vivo experiments whereby xenografts were induced by intracerebral injection of GIC confirmed extended survival (Fig. 6h, left) of mice bearing a GIC19-derived xenograft upon treatment with DSF, with no effect in mice xenografted with GIC31 (Fig. 6h, right).

NTRK2 is a member of the neurotrophic tyrosine receptor kinase (NTRK) family regulating neuronal differentiation, as well as growth and survival (reviewed in ref. [52]). Somatic alterations of this gene have been found in brain tumours including gliomas[53] and the expression of this gene is prognostically relevant in these tumours[54]. Recently, it has been shown to play a key role in the reciprocal signalling between GICs and their differentiated glioblastoma cell progeny[55]. TRK inhibition has emerged as an important therapeutic target in various cancers[56]. We show that *NTRK2* is hypomethylated with a concordant upregulation in patient 30 (Fig. 4b), the latter validated at protein level (Fig. 4c). Treatment with CTX-B, a selective, non-competitive antagonist of NTRK2, known to cross the BBB[57] impacts on the viability of GIC with no effect on the syngeneic iNSC in patient 30 (Fig. 7a, top), while it has no effect on both GIC and iNSC from patient 18, where the molecular signature is not present (Fig. 6a, bottom). CTX-B treatment decreased proliferation of GIC 30, without affecting GIC18 (Fig. 7b); tumour sphere formation (Fig. S4a), cytotoxicity (Fig. S14b), and apoptosis (Fig. S14c) were not affected. Silencing of *NTRK2* (Fig. S14d) inhibits GIC30 proliferation (Fig. S14e) and confirms that the impaired viability observed after CTX-B treatment is NTRK2-dependent as it is lost in GIC upon *NTRK2* silencing (Fig. S14f).

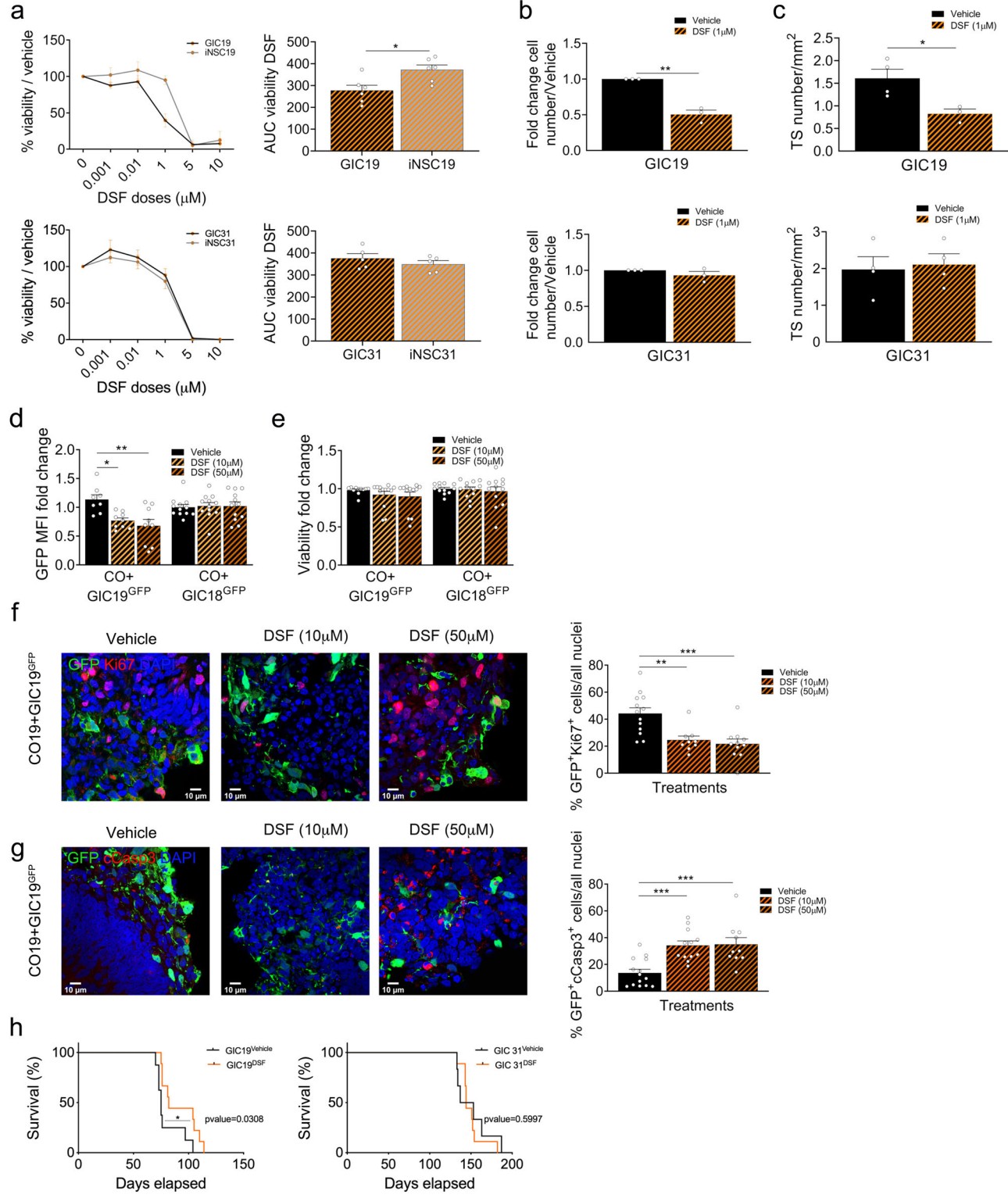

These results were confirmed in the 3D SYNGLICO model where CTX-B treatment impacted the MFI of GIC30$^{GFP+}$ cells within EPSC30-derived CO without affecting viability, indicating loss of GFP+ GIC cells, with no significant effect on the MFI of GFP in GIC18$^{GFP+}$ grown within an EPSC18-derived CO (Fig. 7c, d and Fig. S15a).

In vivo experiments showed extended survival (Fig. 7e, left) in mice bearing GIC30-derived xenograft upon treatment with *tat*-Cyclotraxin-B[57], while it had no effect in mice bearing GIC18-derived xenograft (Fig. 7d, right).

Our results show that deregulation of *ALDH3B1* and *NTRK2* specifically in GIC19 and GIC30 respectively, played a key pathogenetic role specifically in these patients and can be effectively targeted therapeutically in pre-clinical models.

**Absence of overt gastrointestinal toxicity of the compounds at the GIC effective dose as assessed on intestinal organoids.** Drug toxicity on the gastrointestinal tract is a known limiting factor of adjuvant chemotherapy in cancer[58]. Here, we have taken

**Fig. 6 Comparative analysis of transcriptome and methylome of GIC/iNSC identifies ALDH3B1/DSF as patient-specific drug sensitivity. a** Drug treatment of GIC (black curves) and iNSC (grey curves) of patient 19 (top) and 31 (bottom) with doses ranging from 1 nM to 10 μM of DSF (orange dots) or vehicle (black dots). Results are expressed in percentage of viability on the vehicle and were measured at end point, area under the curve (AUC) was calculated from percentages of viability (n = 6 experiment repetitions, two tailed t-test, p value (patient 19) = 0,0155, p value (patient 31) = 0,3459). **b** Proliferation assay of GIC19 (top) and 31 (bottom) treated with vehicle (black bars) or DSF (orange hatched bars) as assessed by cell numbers after 4 days of culture. Results expressed as fold change of cell number at day 4 on day 1 of treatment and standardized on the vehicle treatment conditions (n = 3 experiment repetitions, two tailed t-test, p value (patient 19) = 0,0012, p value (patient 31) = 0,2743). **c** Tumour spheres (TS) were counted after 4 days of treatment of GIC19 (top) and 31 (bottom) with vehicle (black bars) or DSF (orange hatched pattern bars) in non-adherent culture conditions. Results are an average of 10 10× pictures (n = 3 experiment repetitions, two-tailed t-test, p value (patient 19) = 0,0273, p value (patient 31) = 0,7780.). **d, e** Viability of total cells in SYNGLICO (Cerebral Organoids (CO)+GIC$^{GFP}$) assessed as percentages of negative cells for the Zombie NIR$^{TM}$ dye after 4 days of treatment with vehicle or DSF (10 μM, 50 μM) (d) and GFP Mean Fluorescence Intensity (MFI) of GIC$^{GFP+}$ cells (e), (COGIC18: n = 12; GOGIC19 vehicle: n = 8, DSF 10 and 50 μM: n = 9 SYNGLICO per group, one way ANOVA; d) Patient 19: p value (Vehicle vs DSF 10 μM) = 0,0166, p value (Vehicle vs DSF 50 μM) = 0,0029; e) Patient 19: p value (Vehicle vs DSF 50 μM) = 0,4178). Representative GFP and Ki67 (**f**) or cleaved-caspase3 (**g**) immunofluorescence of SYNGLICO after 4 days of treatment with vehicle or DSF (10 μM or 50 μM). The scale bar is 10 μm. Quantification of double positive cells GFP + KI67 (**f**) and GFP + cCASP3 (**g**) is shown on the right (n = 2 SYNGLICOs, GFP + KI67: Vehicle: n = 13 fields, DSF 10 μM: n = 9 fields, DSF 50 μM: n = 11 fields, GFP + cCasp3: Vehicle: n = 14 fields, DSF 10 μM: n = 12 fields, DSF 50 μM: n = 10 fields, one-way ANOVA; **f** p value (Vehicle vs DSF 10 μM) = 0,0033, p value (Vehicle vs DSF 50 μM) = 0,0004; **g** p value (Vehicle vs DSF 10 μM) = 0,0006, p value (Vehicle vs DSF 50 μM) = 0,0007). **h** Survival curve of mice bearing GIC19-derived (left) and GIC31-derived (right) intracerebral xenografts treated with vehicle (GIC19: n = 8, GIC31: n = 5) or DSF (GIC19: n = 10, GIC31: n = 6). Log-rank (Mantel−Cox) test, two-sided, p value (GIC 19) = 0,0450; p value (GIC 31) = 0,8352. All graphs report mean ± SEM. Statistical significance for all panels *p ≤ 0.05, **p ≤ 0.01, ***p ≤ 0.001. Source data are provided in the source data file.

advantage of the availability of patient-matched EPSC to generate intestinal organoids, which have been previously used to assess drug response[59]. Expression of β-CATENIN, E-CADHERIN, VILLIN, and CDX2 confirmed successful differentiation (Fig. S16a). Treatment with PGE1-OH (50 and 100 μM), DSF (10 and 50 μM) or CTX-B (50 and 100 μM) of the intestinal organoids did not show a significant reduction in proliferation or increase in apoptosis, in keeping with lack of measurable toxicity at the doses effective in GIC on a patient-specific basis (Fig. S16b–d).

## Discussion

Here we describe an approach to generate syngeneic neural stem cells and use them as patient-specific comparator to glioma initiating cells for the discovery of pathogenetically relevant mechanisms in glioblastoma and to identify druggable targets in patients.

There is robust published evidence, both in mouse models[60] and in human patients[12], that NSC are the cell of origin of at least a proportion of GBM. Deregulated transcriptional programmes and epigenetic mechanisms have been convincingly shown to play a role, together with genetic lesions, in the neoplastic transformation of these cells into GIC and in conferring essential tumour maintenance properties to the latter[6]. iNSC derived from iPSC have been extensively characterized and shown to share marker expression and functional properties with endogenous NSC[61,62]. Here, we demonstrate that iNSC are remarkably similar to eNSC derived from the brain of the same animal and that they are superior to non-syngeneic eNSC in identifying GBM-defining expression in a mouse model. The strength of our study is the application of this concept to human GBM in a cohort of 10 patients to compare for the first time the epigenetic and transcriptional make-up of GIC with that of patient-matched normal iNSC. This comparison was not feasible to date, as patient-matched endogenous NSC are not surgically accessible and all epigenetic studies in GBM had so far compared epigenetic changes of GBM cells with each other or to comparators obtained from fetal brains[44] or an unrelated donor[55].

Using this syngeneic comparative analysis, we reveal a mechanism of regulatory T cell migration in GBM, mediated via deregulation of Glycosaminoglycans in GIC, in a proportion of GBM patients. A decreased cellular immunity is characteristic of GBM, to which an increased proportion of Tregs among CD4+

TIL (tumour-infiltrating lymphocytes) has been shown to contribute both in patients[63,64] and in pre-clinical models[65,66]. Microenvironment-derived cytokines, such as CCL-2, IDO, and TIM4 (reviewed in[67]) are responsible for the accumulation of Tregs in GBM; however, a role for tumour cells or GIC in modulating Tregs migration is less clear and a link between GAG-GIC and Tregs migration has not been previously described. Sulfation of GAGs plays a regulatory function in modulating extracellular signals in cell−cell and cell−matrix interactions during development and in pathological processes (reviewed in[68]), such as cancer. Chondroitin sulfate proteoglycans (CSPGs) and their GAG side chains are key constituents of the brain extracellular matrix (ECM) implicated in promoting GBM invasion[69]. Changes in GAG sulfation have been proposed as cancer biomarkers, however, person-to-person variation in GAG composition has hampered their applicability[70]. Notably, our results show that GIC regulate Tregs migration specifically through a CS/DS-dependent mechanism both in vitro and in vivo and suggest that this is mediated by their altered sulfation both in secreted and membrane GAG from GIC. Therapeutic approaches aiming at modulating GAG sulfation, for example, to deplete infiltrating Tregs, may therefore require this type of patient-specific approach and the value of a FoxP3/CD4 index as a predictive biomarker in surgical pathology should be further assessed.

Importantly, we provide proof of principle that combined comparative analysis of the methylome and transcriptome of GIC and matched iNSC identifies druggable targets in a patient-specific fashion in a proportion of cases. The ALDH inhibitor disulfiram, which has been trialled clinically in GBM patients with unmethylated MGMT because of its MGMT inhibitory role[71], may be more effective or have additional effects on patients with ALDH3B1-specific deregulation. Hypomethylation/high expression of NTRK2, a druggable tyrosine receptor kinase[56] playing a role in the reciprocal signalling between GIC and their differentiated glioblastoma cell progeny[55], predicts response to the specific inhibitor cyclotraxin-B. Finally, we provide compelling evidence that PTGER4 agonists are effective in a patient-specific fashion predicted by hypermethylation/low expression of the gene in the syngeneic GIC/iNSC comparison in 2D cultures and in a 3D syngeneic GLICO model. Development of compounds that can be administered in vivo will be required for further pre-clinical assessment of its applicability and assessment

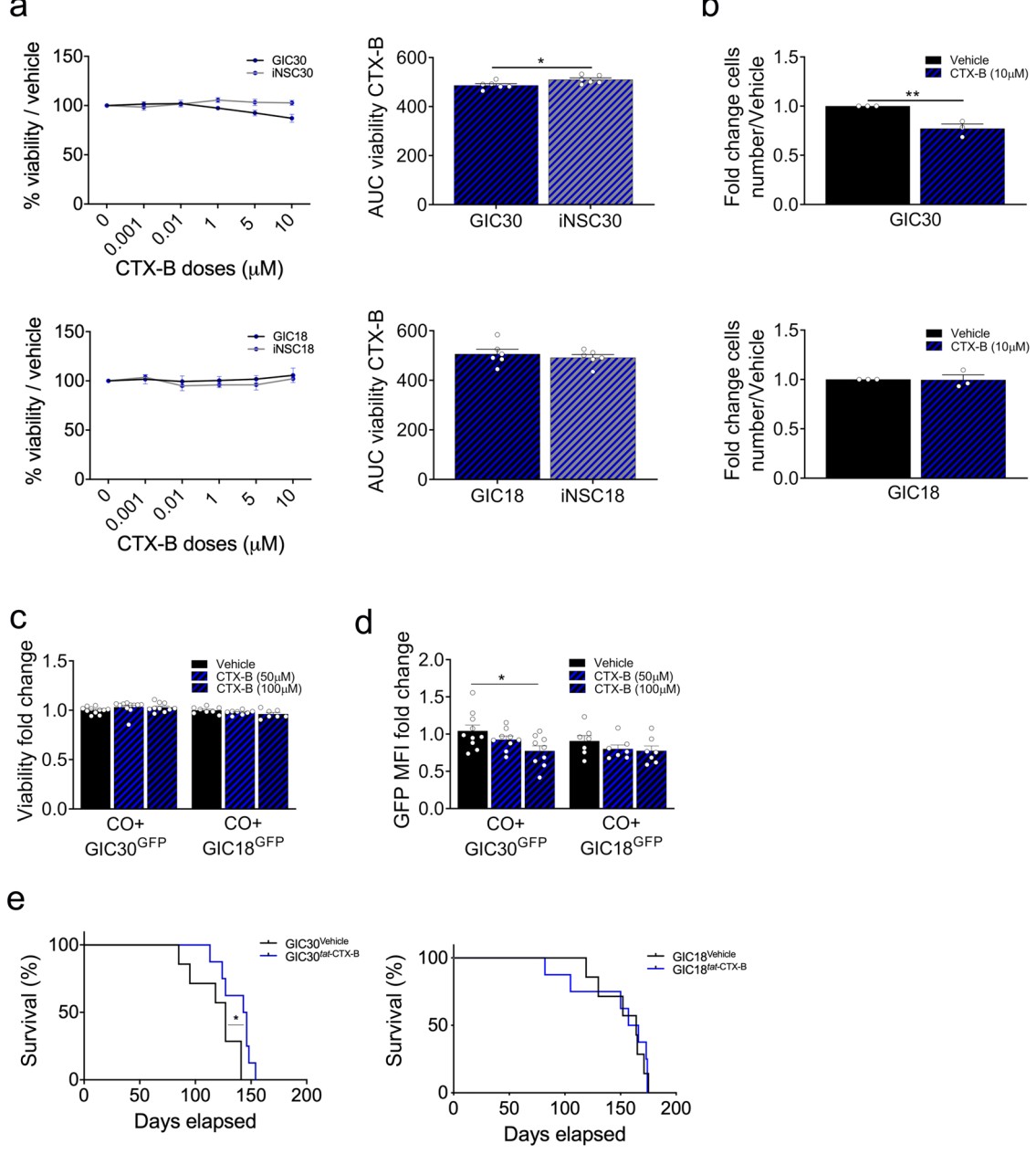

**Fig. 7 Comparative analysis of transcriptome and methylome of GIC/iNSC identifies NTRK2/CTX as patient-specific drug sensitivity. a** Drug treatment of GIC (black curves) and iNSC (grey curves) of patient 30 (top) and 18 (bottom) with doses ranging from 1 nM to 10 μM of CTXB (blue dots) or vehicle (black dots). Results are expressed in percentage of viability on the vehicle and were measured at end point, area under the curve (AUC) was calculated from percentages of viability ($n = 6$ experiment repetitions, two tailed $t$-test, $p$ value (patient 30) = 0.0326, $p$ value (patient 18) = 0.5394). **b** Proliferation assay of GIC30 (top) and 18 (bottom) treated with vehicle (black bars) or CTX-B (blue hatched bars) assessed by cell numbers after 4 days of culture. Results are expressed as fold change of cell number at day 4 on day 1 of treatment and standardized on the vehicle treatment conditions ($n = 3$ experiment repetitions, two-tailed $t$-test, $p$ value (GIC 30) = 0,0080, $p$ value (GIC 18) = 0.9363). Viability of total cells of SYNGLICO (cerebral organoids (CO) + GIC[GFP]) after 4 days of treatment with vehicle or CTX-B (50 μM, 100 μM) (**c**) and GFP Mean Fluorescence Intensity (MFI) of GIC[GFP+] cells (**d**) (COGIC30 vehicle: $n = 10$, CTX-B 50 and 100 μM: $n = 9$; COGIC18: $n = 7$ SYNGLICOs per group, one way ANOVA, two-sided. **c** Patient 30: $p$ value (Vehicle vs CTX-B 100 μM = 0.3759); **d** Patient 30: $p$ value (Vehicle vs CTX-B 100 μM = 0.0230). **e** Survival curve of mice bearing GIC30-derived (left) and GIC18-derived (right) intracerebral xenograft treated with vehicle ($n = 7$) or tat-Cyclotraxin-B ($n = 8$), log-rank (Mantel−Cox), two-sided, $p$ value (GIC 30) = 0.0290. All graphs report mean ± SEM. Statistical significance for all panels *$p \leq 0.05$, **$p \leq 0.01$, ***$p \leq 0.001$. Source data are provided in the source data file.

of its potential clinical value. Importantly, the epigenetic regulation of *PTGER4* gene expression by DNA methylation could represent a valuable biomarker to be further developed as a predictor of drug response, particularly taking into account the roll out of methylation-based classification approaches[18] in brain tumours.

More work will be required to assess whether the SYNGN (syngeneic comparison of GIC and iNSC) pipeline can be used in the clinical management of patients with GBM in a timeframe suitable for adjuvant treatment and whether syngeneic GLICO and intestinal organoids accurately predict tumour response and non-neoplastic cell toxicity on a patient-specific basis (Fig. 8). It

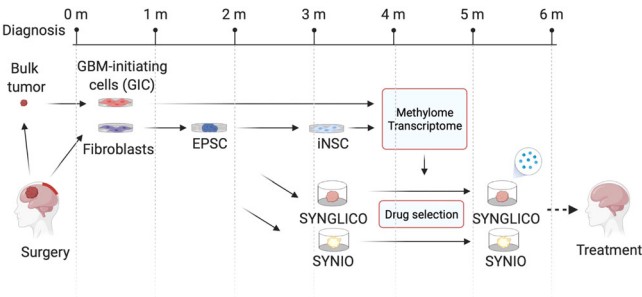

**Fig. 8 Schematic representation of the SYNGN pipeline.** Timeline from GBM diagnosis to potential patient treatment with the SYNGN pipeline. SYNGLICO syngeneic glioblastoma cerebral organoids. SYNIO syngeneic intestinal organoids.

will be important to assess whether applying drug sensitivity prediction algorithms[72] with a syngeneic comparator included, may increase the number of cases for which a matching drug can be identified and/or advance our understanding of whether our pipeline can be applied at recurrence and whether it is suitable to predict response to combination therapies. Nevertheless, our approach has identified functionally relevant epigenetic modifications of target genes that predict differential response to drugs currently in clinical use, which could be trialled for GBM treatment.

## Methods

**Human cell cultures**. Informed consent was obtained and ethical approval was available for the study (National Health Service, Health Research Authority, National Research Ethics Service 08/H0716/16 Amendment 1 17/10/2014). GIC were isolated from bulk tumour following a published protocol[21]. In brief, fresh GBM tissue was sliced and triturated with a razor blade, dissociated with Accumax (Sigma, A7089) at 37 °C for 10 min then filtered through a 70 μm cell strainer. Dissociated cells were plated on laminin-coated 6-well plate in NeuroCult NS-A Proliferation kit media (STEMCELL, 05751), heparin (2 μg/ml; Gibco 12587-010), mEGF (20 ng/ml, Prepro Tech, 315-09) and hFGF (10 ng/ml; Prepro tech, AF-100-18B). Established cells were passaged when 70% confluent, frozen in Stem Cell Banker (Ambsio ZENOAQ, 11890), and stored in liquid nitrogen. Cell lines used in this study are primary lines derived from human tumours, they have been characterized by transcriptomic profiling and cultured according to current practice, including contamination assessment.

Fibroblasts cultures were established from small strips of dura mater, collected at the surgery. The tissue was minced with a scalpel then spun down and resuspended in trypsin for 5 min at 37°. To stop the reaction fresh fibroblast media (DMEM, Glutamax, 10% Foetal calf serum, 2% L-Glutamine, and 1% penicillin-streptomycin) was added. Samples were centrifuged and resuspended in fresh media, then plated in six well plates (Corning #BC010). Media was topped up frequently during the first week then changed every other day.

Fibroblasts reprogramming was performed following a published protocol[23]. Briefly, fibroblasts were electroporated with reprogramming vectors expressing Oct4, c-Myc, Klf4, Sox2 (OCKS 4F, 5 μg) and Rarg, Lfh1 (RL 2F, 5.0 μg) using Amaxa Nucleofector (Lonza, Germany) then plated on SNL feeders with M15 media (Knockout DMEM Invitrogen, 15% Fetal Bovine Serum Hyclone, 1X Glutatamin-Penicillin-Streptomycin Invitrogen, 1X non-essential amino acids Invitrogen). When colonies appeared, media was replaced with EPSCM (DMEM/F12 Invitrogen, 20% Knockout Serum Replacement Invitrogen, 1X Glutamin-Penicillin-Streptomycin, 1X non-essential amino acids Invitrogen, 0.1 mM β Mercapto-ethanol Sigma, 106 U/ml hLIF Millipore supplemented with the following inhibitors: CHI99021 Tocris 1 μM, JNK Inhibitor VIII Tocris 4 μM, SB203580 Tocris 10 μM, A-419259 Santa Cruz 1 μM and XAV939 Stratech 1 μM). Colonies were picked and plated in 24 well SNL feeders plates for expansion and characterization.

For embryoid bodies (EB) generation, EPSC lines were harvested and feeders removal was performed in T25 flasks for 30 min at 37° to allow feeders to attach to the bottom. Floating EPSC harvested were transferred in ultra-low attachment 96 well plates at different densities (60, 45, 35, and 20 K) in EB media (DMEM/Knockout Invitrogen #10829-018, Fetal Bovine serum Gibco #16141061, non-essential amino acid Invitrogen #11140, glut-pen-strep Invitrogen #10378 and β mercapto-ethanol) changed every other day. After 7 days, they were transferred in 24 well plates on gelatine coated glass coverslips and after 10−14 days, cells were fixed with PFA 4%, 30 min at room temperature, and immunocytochemistry was performed.

For iNSC induction, a commercially available kit was used following the manufacturer's protocol (Gibco, #A1647801). Briefly, EPSC colonies were harvested, feeders' removal was performed and EPSC plated at density from 0.25 to $0.75 \times 10^6$ per well of six well geltrex coated plates in EPSCM and Rock inhibitor 10 μM (Stem cell technology #Y27632). The next day, the media was replaced with Gibco neural induction media (Neurobasal media #211030, pen-strep 1X and neural induction supplement 1X #A1647801, rock inhibitor 10 μM), then changed every 2 days until day 7, cells were passaged at the density of $1 \times 10^5/cm^2$ in Gibco neural expansion media (Neurobasal 0.5X, Advanced DMEMF/12 0.5X #12634, pen-strep 1X and neural induction supplement 1X, rock inhibitor 10 μM), then media was changed every 2 days, cells were dissociated with accutase (Millipore #SCR005) and replated 2–3 times to establish the iNSC lines. Cells were frozen in synth-a-Freeze cryopreservation medium (Gibco #A12542). GIBCO® Human Neural Stem Cells (H9 hESC-Derived, #N7800-100) were used as control, cultured in StemPro® NSC SFM (Cat. no. A10509-01).

For differentiation of iNSC along the astrocytic and neuronal lineages, iNSC cells were plated on geltrex (Gibco #A1413302) coated culture dishes. After 2 days, the media was changed to neural differentiation media (Neurobasal medium Gibco #21103, B27 serum-free supplement 1% Gibco #17504) and changed every other day until day 10. iNSC differentiation along the oligodendrocytic lineage was performed following the protocol developed by[73] with some adaptations. Briefly, at D0, neural expansion media was changed to N2 media made of 47.5 ml of Advanced DMEM/F12 (Invitrogen, #11320033), 0.5 ml of non-essential amino acids (Invitrogen, #11140035), and GlutaMAX (Invitrogen, 35050038), 50 μl of β-mercaptoethanol, 0.5 ml of N2 supplement (Invitrogen, #17502048) changed daily with fresh 100 nM Retinoic acid (RA) (Sigma, #R2625) and+ 1 μM smoothened Agonist (SAG) (Millipore, #566660) until day 4, then replaced by N2B27 media made of N2 media with 1 ml of B27 supplement (Invitrogen, #12587010) and 114 μl of insulin (Sigma, #I9278) changed daily with fresh RA and SAG. Cells were then scraped, and the single cells suspension was transfered to an ultra-low attachment 24-well plate (Corning Cat. #3473) in N2B27 media and changed every other day for 10 days. On day 22, cell aggregates were plated onto Poly-L-Ornithine and Laminin coated plates and cultured in adherent conditions for 45 days with PDGF media made of N2B27 media with 5 μl of PDFR-AA (R&D, #221-AA1050), 2.5 μl of IGF (R&D, #291-G1-200), 25 μl of HGF (R7D, #294-HG-025), 5 μl of NT3 (Millipore, #GF031), T3 (Sigma, #T2877), Biotin (Sigma, #B4639) and cAMP (Sigma, #D0260) changed every other day.

Cells were then fixed with 4% PFA for 30 min at room temperature then immunostained with primary antibodies against GFAP, TUJ1, and Olig2 (Table S3).

**Karyotyping**. In order to obtain metaphase chromosomes, iNSC were grown to 70% confluency and the media changed a day before harvesting. The cells were then treated with colcemid (0.04 μg/ml) for 1 h, followed by hypotonic treatment in 0.4% potassium chloride for 1−4 min at 37° and fixation in 3:1 methanol-glacial acetic acid. Slides were air-dried and stored desiccated at 4 °C until required. For chromosome analysis, slides were counterstained with 0.5 μg/ml DAPI (4, 6-diamidino-2-phenylindole) and mounted in Citifluor. Images were captured using a Zeiss Axiophot microscope equipped for epifluorescence using a Zeiss plan-neofluar 100× objective and an optivar set at 1.0 × magnification. Separate grey-scale images were recorded with a cooled CCD-camera (Hamamatsu, Welwyn Garden City, UK). Image analysis was performed using SmartCapture X software (Digital Scientific, Cambridge, UK), followed by chromosome counting using ImageJ 1.50i. 10−30 Metaphase spreads were captured for each culture and chromosomes counted, 3−5 metaphase spreads were then analyzed in detail, and karyotypes confirmed as either 46, XY, or 46, XX without any gross rearrangement.

**Generation and characterization of syngeneic GLICO**
*Transition of feeder-dependent EPSC to feeder-independent EPSC (feeder-free).* EPSC colonies were plated in geltrex coated 24 well plate (1 colony/well) in EPSC feeder-dependant conditioned media and mTeSR Plus media (1:1) (Stem cell technology, #100-0276) and Rock Inhibitor for 24 h then media was change to mTeSR. Cells were passage using 0.5 mM EDTA for expansion.

*Cerebral organoid generation.* Cerebral organoids were generated and cultured as per manufacturer's instructions (StemCell Technologies, #05825) with minor adaptations. Briefly, for EB generation, feeder-free EPSC were dissociated into single cells with gentle cell dissociation reagent (StemCell Technologies, # 100-0485), and 90,000 cells were plated into each well of a 96-well round-bottom ultra-low attachment plate (Corning) in seeding media (EB formation media+50 μM ROCK Inhibitor). EBs were fed every 2 days for 7 days (EB diameter ≥ 300 μm), and then transferred to a 24-well ultra-low attachment plate (Corning) in a neural induction medium for 3 days. EBs were then embedded in Matrigel hESC-Qualified Matrix (Corning, #354277) droplets and transferred into a six-well ultra-low attachment plate (Corning) in a neural expansion medium for 3 days. After 3 days, the expansion medium was removed from the wells and replaced with a maturation medium. The plate was transferred on an orbital shaker at 37 °C at 90 rpm.

*Co-culture with patient-derived GFP-labelled GIC.* GIC were transduced with pLoxGFP (Addgene, #12241) as below (see lentiviral production). On day 1 of co-culture, 1-month old cerebral organoids derived from the patient-specific EPSC were transferred into each well of a 96-well round-bottom ultra-low attachment plate. The excess medium was removed and 50,000 GFP-expressing GIC were plated in each organoid-containing well in GIC medium. On day 2 of co-culture, the excess medium was removed from each organoid-containing well and an additional 50,000 GFP-expressing GIC were added in fresh GIC media, and plates were incubated overnight. On day 3, each tumour-bearing organoid was transferred into a well of a 12 well super-low attachment Nunclon Sphera plate (Thermo Scientific) and maintained in cerebral organoid maturation media on an orbital shaker at 130 rpm for 5–7 days followed by 4-day drug treatment.

**Generation and characterization of EPSC-derived intestinal organoids.**
Intestinal organoids were generated and cultured as per the manufacturer's instructions (StemCell Technologies, #05140). Briefly, feeder-free EPSC were cultured in matrigel coated 24 well plates and once they reached confluence, endoderm differentiation was induced for 3 days, followed by 3−7 days of definitive endoderm differentiation until formation and release of spheroids from the monolayer. Mid/hindgut obtained were then embedded in a dome of matrigel in a new Nunclon Delta surface-treated 24 well plate with flat bottom and cultured in an intestinal organoid medium, changed every 3–4 days. Organoids were passaged once a week at the density of 50 organoids/matrigel dome.

**Animal procedures and murine cells cultures.** All procedures were performed in accordance with licences held under the UK Animals (Scientific Procedures) Act 1986 and later modifications and conforming to all relevant guidelines and regulations.

*Cell cultures and assays.* Primary endogenous NSC (eNSC) were isolated from the sub ventricular zone (SVZ) of 3 months old C57Black6/J mice following established protocols[39]. Briefly, SVZ were dissected from the cerebral hemispheres and digested using Papain dissociation kit (PDS, Worthington #0031150 and #003153), plated at a density of $4 \times 10^4$ mL$^{-1}$, and grown as Neurospheres (NS) for 4 days in DMEM/F12 (Invitrogen #31330), mouse recombinant EGF and human recombinant FGF (20 ng/ml, Peprotech #31509 and #100-18B), 2% B27 (Invitrogen #12587) and Pen/Strep 1X (Invitrogen #15140), then media was changed every 4 days. After the first passage, cells were plated in the adherent condition in geltrex (Gibco # A1413302) coated six well plates (corning #3516) at a density of $0.5 \times 10^6$ cells. Dissociation between passages was performed enzymatically with accutase (Millipore #SCR005). After 2 passages, media was changed to neural expansion media (Gibco Advanced DMEM/F 12 0,5X #12634, Neurobasal medium 0,5X #21103, 1% Pen-Strep and 2% neural induction supplement #A1647801). Dura mater was collected from C57Black6/J mice before dissection of the hemispheres for eNSC isolation. Fibroblasts isolation, EPSC reprogramming, and EB formation were carried out following the same protocol described above for the human cells. EPSC were then induced into iNSC following two protocols: the Gibco protocol (iNSC$^{Gibco}$) described above for the human cells and a bespoke protocol[32]. For the latter, feeders were removed and EPSC colonies dissociated with accutase and plated in geltrex coated six well plates in EPSCM and rock inhibitor 10 µM at density from 0.25 to $0.5 \times 10^6$ cells/well. Media was changed the next day with N2B27 induction media (Neurobasal medium Gibco #12348017, DMEM/F21 Gibco #11330032, N2 supplement #17502048, B27 supplement 12587010, EGF and bFGF 10 ng/ml peprotech and Pen-Strep-Glut Sigma) and afterwards every 4 days for a week. Cells were passaged with accutase and plated at density of $0.5 \times 10^6$ cells/well in geltrex coated six well plates and media was replaced by N2 expansion medium (DMEM/F21 Gibco #11330032, N2 supplement #17502048, EGF, bFDF, and Pen-Strep-Glut) and changed every 2 days, passages were performed by enzymatic cells dissociation with accutase 2−3 times to establish iNSC N2B27 lines (iNSC$^{N2B27}$). eNSC, iNSC$^{Gibco}$, and iNSC$^{N2B27}$ were frozen in synth-a-Freeze cryopreservation medium (Gibco #A12542).

NS assay was performed by plating endogenous and both induced NSC as single cells in ultra-low attachment 96 well plates for 96 h, pictures were taken with EVOS XL Core microscope.

*Orthotopic xenografts.* Eight-to twelve-week-old NOD SCID CB17-Prkdcscid/J mice (purchased from The Jackson Laboratory) were anaesthetized with isoflurane gas and $5 \times 10^5$ cells (hGIC or hiNSC) in 10 µL PBS were slowly injected with a 26 gauge Hamilton syringe needle into the right cerebral hemisphere with the following coordinates from the bregma suture: 2 mm posterior, 2 mm lateral, 4 mm deep, 10° angle. The scalps were sutured with 4-0 Coated Vicryl Suture (Ethicon) and mice were allowed to recover from the surgery on a heat-map until they were fully awake. Post-operative checks were performed twice a day for five days, then once a day and body weight was monitored once a week. For survival experiments, mice were kept on tumour watch until symptoms developed. Mice were euthanized by neck dislocation and brains were harvested for histology and immunohistochemistry.

For hiNSC xenografts, mice did not show any neurological sign and the experiment was terminated 4 months post injection by neck dislocation. Four

months was chosen as an end point as it was the average survival of the mice xenograft with the 10 GIC lines.

Subcutaneous xenografts: eight to twelve-week-old NOD SCID CB17-Prkdcscid/J mice (purchased from The Jackson Laboratory) were anaesthetized with isoflurane gas and $4 \times 10^6$ GIC cells in 100 µL of 50% matrigel were injected with an insulin syringe at four sites (two flanks and behind both ears). When tumours were established, they were injected with 25U of Chondroitinase ABC (chABC)[74] or vehicle in two different sites of the tumour. After 24 h, $10 \times 10^6$ human Tregs were injected intravenously after being isolated from PBMC derived from frozen human buffy coats (REC 17/LO/1061). Briefly, CD4 T cells were enriched in PBMC with Rosette Sep enrichment cocktail (StemCell, 15022) and CD25 positive cells were selected with the microbeads from Mylteni (130-092-983). Twenty-four hours following the Tregs injections, mice were euthanized and tumours were collected, embedded in OCT, and frozen for histological assessment.

**Sequencing.** RNA and DNA were extracted from cell pellets using the RNA/DNA/Protein Purification Plus kit (NORGEN, #47700), following the manufactured protocol. Mice and human RNA samples were sequenced on HiSeq4000, at 75 PE after polyA selection, by Oxford Genomics Centre. Human DNA methylation was assessed on the Illumina Infinium MethylationEpic kit at the UCL Genomics, Institute of Child Health, and mouse DNA methylation was performed by Reduced Representation Bisulfite Sequencing (RRBS). For RRBS library preparation DNA samples were digested overnight with restriction enzyme Mssp1 (Agilent), and then prepared following the manufactured protocol of NEBNext® Ultra™ II DNA Library Prep Kit for Illumina® (Biolab, E7645) and bisulfite conversion following Zimo kit (Qiagen, D5001) and run on NextSeq 500 Mid Output Run 150 cycles (Genome Centre, Barts and the London School of Medicine and Dentistry).

**Tregs migration assay.** PBMCs were isolated from the blood of healthy donors (REC 17/LO/1061). They were collected between the top plasma layer and the bottom ficoll medium of a Histopaque1077 (Sigma-Aldrich) gradient and resuspended in PBS, washed and plated in a petri dish in RPMI 1640 + 10% FBS incubated 45 min to deplete macrophages. The supernatant was then harvested and PBMC were counted and frozen in FBS 90% and DMSO 10%. For ChABC (Sigma) treatment, the compound was added to cell culture media at the final concentration of 0.01U/ml for 48 h. Then cells were stained for chondroitin 56 (Abcam) or harvested to be plated in transwell chamber.

Conditioned-media from GICs or iNSCs or cells (20 K) were added to the geltrex coated top chamber of a transwell 5 µm pore (Sarstedt, 83.3932.500) on a 24 wells plate. Twenty-four hours later, PBMCs (300 K cells) in RPMI with FBS 10%, were added to the top chamber for a 4 h incubation at 37 C, CO$_2$ 5%. Cells were harvested in the bottom chamber, counted, and stained by flow cytometry.

**Analysis of GAG composition using AMAC-labelling and RP-HPLC.** Cultured GICs and iNSCs were washed with PBS then lysed using 1% (vol/vol) Triton X-100 for 2 h at room temperature. Conditioned-media and cell extract samples were treated with 100 µg/mL Pronase (Roche Diagnostics) for 4 h at 37 °C. Cell extracts were incubated for 15 min at 90 °C to deactivate the Pronase, before treating with 14 µg/mL DNase I (Sigma) and 10 mM MgCl$_2$ overnight at 37 °C to prevent DNA interference with downstream processing. All preparations were loaded onto 1.5 mL diethylaminoethyl (DEAE)-Sephacel columns (Sigma) and washed with 50 mL of 0.25 M NaCl, 20 mM NaH$_2$PO$_4$.H$_2$O (pH 7.0) to remove hyaluronan. The sulfated GAGs were eluted with 5 mL of 1.5 M NaCl, 20 mM NaH$_2$PO$_4$.H$_2$O (pH 7.0), and desalted using PD-10 Sephadex G-25M pre-packed columns (GE Healthcare). Purified GAGs were freeze-dried, then digested with GAG lyases specific for either HS or CS/DS for 12 h at room temperature. HS chains were digested using 0.8 mIU of each heparinase I, II, and III (from Flavobacterium heparinum, Iduron) in 100 µL of 0.1 M sodium acetate, 0.1 mM calcium acetate (pH 7.0). CS/DS chains were digested using 0.8 mIU chondroitinase ABC (chABC) (Sigma) in 100 µL of 50 mM Tris, 50 mM NaCl (pH 7.9). The resulting disaccharides were freeze-dried then re-suspended in 10 µL of 0.1 M AMAC and incubated at room temperature for 20 min. To each reaction, 10 µL of 1 M NaBH$_3$CN was added and incubated at room temperature overnight in the dark. The AMAC-labelled samples were separated by RP-HPLC in duplicate using a Zorbax Eclipse XDB-C18 RP-HPLC column (35 µM, 2.1 × 150 mm) (Agilent Technologies), following published protocols[75,76]. Disaccharides were detected by fluorescence and were identified and quantified in comparison with known amounts of commercial disaccharide standards (Iduron).

**Immunofluorescence and immunohistochemistry**
*Staining.* EPSC, NSC, and EB in adherent cultures on a glass coverslip were washed once with PBS then fixed with PFA4% for 30 min at room temperature. After three washes with PBS, blocking with 3% Donkey normal serum, 0.3% Triton X100 PBS for 30 min at room temperature was performed prior to incubation with primary antibodies (see Table S3) at 4° overnight. After three washes with PBS and 1 h incubation with secondary antibodies (see Table S3) diluted in PBS, cells were washed again with PBS, and coverslip were mounted on SUPERFROST slides with mounting media containing DAPI for nuclear counterstaining (Vectashield H-1500). Microscope analysis was performed on IN Cell Analyzer 2200 Cell Imaging

System (GE HealthCare life sciences) for the EB staining or Leica DM5000 Epi-Fluorescence microscope for NSC and EPSC.

Cerebral and intestinal organoids were fixed in PFA (4%), overnight and 30 min at room temperature, respectively. After washes with PBS, organoids were incubated in 30% sucrose until they sunk, embedded in 7.5% gelatin (in 10% sucrose), and then frozen in cold isopentane. Cerebral organoids and GLICO were cryosectioned at 20–25 µm and intestinal organoids at 10 µm using the Leica cryostat. For staining, slides were rehydrated, permeabilized for 1 h at RT (4% serum, 0.25% Triton-X in PBS), then incubated in normal serum (4% serum, 0.1% triton-X in PBS) and the relevant primary antibody overnight at 4°. Primary antibodies for Sox2, Nestin, and NeuN (cerebral organoids) and Villin, E-Cadherin, β-catenin, and CDX2 (intestinal organoids), as well as Ki67 and cleaved-caspase3, were used (Table S3). Secondary antibodies were incubated for 2 h at room temperature, and slides were mounted with mounting media containing DAPI.

*Quantification.* Chondroitin sulfate fluorescence staining was analyzed by CTCF (corrected total cell fluorescence): CTCF = Integrated Density − (Area of selected cell × Mean fluorescence of background readings) with Image J software.

Subcutaneous tumour injected in vivo with Tregs GFP-labelled slides were scanned with TISSUEFAX® and GFP positive cells and nuclei were automatically counted with Strataquest software.

Cryosections of the organoids were stained with GFP and Ki67 or GFP and Cleaved-caspase-3, and nuclear counterstained with DAPI as described above. For intestinal organoids, mosaic of all organoids area were captured at high magnification, nuclei and positive cells were then manually counted. For cerebral organoids, 4–6 fields per condition for each individual biological sample were captured as Z-stacks of approximately 10um at 63x magnification using the Zeiss 880 Laser Scanning Confocal Microscope with Fast Airyscan and Multiphoton. Z-stack images were then Airyscan processed and analyzed using the IMARIS software. Briefly, each image was converted to a.ims file, each channel (GFP, RFP, DAPI) was set to a threshold in order to use the spot function analysis and applied to all images. GFP spots represent GIC, RFP spots Ki67 or Cleaved-caspase-3, and DAPI spots as cells (nuclei). Spot counts and double-positive overlapping spots results were then exported and analyzed in Graphpad Prism 8.

## Flow Cytometry

*Transwell assay.* Cells were washed once with PBS then plated and pelleted in 96 wells plate in 200 µl PBS. Antibodies CD4-PE, CD8-AF700, CD25-PECy7, CD127-APC, CD56-FITC CD56-FITC, CD14 Percp Cy5.5, and L/D marker (see Table S3) were diluted in FACS buffer and incubated at room temperature for 30 min. Cells were then washed, permeabilized, and fixed with Transcription Factor Staining Buffer Set (eBiolegend 00-5523-00). Foxp3 antibody was diluted in the permeabilization buffer and incubated for 20 min at room temperature. Finally, cells were washed, resuspended in FACS Buffer and run on BD LSRII device, with a FACSDiva software Version 8. Data were transferred and analyzed using the FlowJo software V10 (Tree Star, Oregon, USA).

The gating strategy employed was as followed—cells were first selected based on size using the FSC-A and SSC-A. Then live cells were selected for by gating on cells negative for the fixable viability dye-e506. Gating for T cells employed CD8 and CD4 markers. Gating for Tregs population relied on $CD4^+Foxp3^+CD25^+CD127^-$ marker expression, monocytes relied on $CD14^+$ population, natural killer on $CD56^+$ population while other immune T cells were relied on $CD4^+Foxp3^-CD25^-$ and $CD8^+Foxp3^-$.

*CO-GIC cultures.* Upon dissociation with the Neural Dissociation kit and staining with the Zombie NIR™ Fixable Viability Kit, all samples were analyzed using the BD LSRII flow cytometer. After exclusion of cell debris (FACS plots SSC-A/FSC-A), and gating of single cells (SSC-W/FSC-A), the overall viability of the organoid was assessed after gating for Zombie NIRTM negative cells (live cells), shown on plots as a histogram of Comp-APC-Cy7-A. From the single cell gating, the overall GFP+ population was detected (SSC-A/Comp-FITC-A) including both live and dead GFP-expressing GIC, from which the MFI was measured as the geometric mean of Comp-FITC-A. Within the live cell gating, cells were gated for their GFP expression, from which the MFI was calculated as described before. For co-cultures with epigenetically edited GIC18, carrying the BFP/mCherry screening markers, samples were analyzed and gated in the same manner to assess viability. In order to assess the MFI of GIC, double-positive cells for BFP and mCherry were gated, from which the MFI of BFP was analyzed.

All FACS analysis was done using FlowJo v10.6.2. All treated co-cultures' viability and GFP MFI were normalized to the vehicle control of each line. Each organoid was considered an independent biological replicate. All results of biological replicates were analyzed using Oneway-ANOVA in GraphPad Prism 8.

## Drug treatment

In vitro drug treatments in 2D cultures were performed on 2k cells plated in 96 well plates; 4 days treatment was performed with a range of doses from 1 nM to 10 µM or vehicle. At end-point, cell viability and cytotoxicity were measured with CellTiter-Glo Luminescent Cell Viability Assay with CellTox Green Cytotoxicity Assay (Promega kits G7570 and G8741).

*Drug treatment on co-culture cerebral and intestinal organoids.* Syngeneic GLICO were allowed to develop for 5–7 days before initiating the treatment with 50 µM and 100 µM PGE1-OH, 50 µM and 100 µM CTX-B or 10 µM and 50 µM DSF. Each drug was added to the final volume on day 1, followed by drug top-up every day for the following 3 days, without discarding any of the contents in each well. Vehicle control co-cultured organoids were treated with DMSO. Organoids were then processed and analyzed with the BD LSRII cytometer as described above or fixed and processed for immunohistochemistry. For FACS analysis on SYNGLICO model: On day 4 post-treatment, all contents of each well including the co-cultured cerebral organoids were collected into tubes and the Neural Dissociation kit (MACS Miltenyi Biotec) was used to dissociate organoids into single-cell suspension.

Drug treatment in xenograft mice model started two months after intra-cranial injection of GIC, 5 times a week in IP with DSF (100 mg/kg) or vehicle (DMSO 5%)[51] and *tat*-Cyclotraxin B (BIO S&T, 20 mg/kg) of vehicle (saline solution) (Cazorla et al., 2010). Mice were weighed 3 times a week to adjust doses.

## Plasmid design and construction

FUW-dCas9-Tet1-CD construct plasmids and the pgRNA (empty-scaffold plasmid guide RNA) were a gift from Dr. Rudolf Jaenisch[46,45]. All *PTGER4* gRNA plasmids were designed based on the sequence near the hypermethylated CpG islands identified in syngeneic comparison (Fig. 4b) (CpG1: cg10603233, CpG2: cg24716010, CpG3: cg11408333, CpG4: cg06013215, CpG5: cg03337243, CpG6: cg17421097, CpG7: cg02026948, CpG8: cg04727116, CpG9: cg21149775, CpG10: cg01952088, CpG11: cg01897756, CpG12: cg04126707). Six guides were designed using the Ensembl GRCh37 reference. In order to clone the target sequences into the pgRNA, two complement oligonucleotides were designed (see Table S3). To ensure cloning compatibility, TTGG- and AAAC- were added to the oligonucleotides sequence (5′-3′). Oligos were designed to be cloned in the guide RNA plasmid to specifically fused proteins in a locus-specific manner.

Oligonucleotides duplexes were formed to establish single guide RNA (sgRNA), by oligonucleotide phosphorylation using the T4 Polynucleotide Kinase (PNK, ThermoFisher). 100 µM of each gRNA sequence (forward and reverse) with 25 mM ATP and PNK/Reaction buffer (10×) were added in a PCR tube, followed by 30 min incubation at 37 °C, 5 min at 95 °C, and ramp to 4 °C by 0.1/s to join the single C at the 3′ end of the reverse oligo annealed the sequence to the additional G added to the forward oligo. The pgRNA vector was cut and modified with the restriction AarI enzyme (ThermoFisher) targeting the AarI sites (CACCTGC) at 37 °C incubation overnight. The enzyme was then inactivated at 65 °C for 20 min. The gRNA plasmids were cloned by inserting annealed oligonucleotides into AarI site modified pgRNA plasmid. After ligation of the annealed duplexes into the digested vector, the cut plasmid was added along with the duplex oligos, ligation buffer, and quick ligase (Quick Ligation kit, NewEngland Biolabs) for 5 min at 25 °C. To ensure only circularized DNA was present, the ligation product was digested with plasmid-safe ATP-dependent DNase at 37 °C for 30 min to digest any uncircularised DNA.

## Lentiviral production

For the production of lentiviruses expressing the two FUW construct plasmids dCas9-Tet1CD (catalytic domain), and all gRNA plasmids, pLoxGFP plasmid (Addgene), plasmid for overexpression of *PTGER4* (Horizon-Dharmacon) and sh*NTRK2* plasmid (Horizon-Dharmacon), $5 \times 10^5/cm^2$ HEK 293T cells were transfected with lipofectamin 3000 (Invitrogen, L3000), the lentiviral plasmids pCMV-G, pCMV-HIV1 (Table S3) and the plasmid of interest. After 48 h in culture, the supernatant was collected, cell debris was removed by filtration (0.45 PVDF filter, Sartorius) and the lentiviral particles were precipitated with polyethylene glycol (Sigma Aldrich) and stored at 4 °C for 16 h. Lentiviral particles were concentrated by centrifugation (30 min, $1500 \times g$, 4 °C), resuspended in sterile PBS, and stored at −80 °C. Transducing units were then determined by plating $0.5 \times 10^5$ HEK 293T cells into 12-well dishes and exposing them to serial dilutions $10^{-1}$ to $10^{-5}$ of the lentiviral supernatant. Cells were collected 96 h from the transduction and the percentage of GFP- positive cells was determined by FACS (BD FACS Canto II). The titration in transducing units per ml (TU/ml) was calculated according to the following formula: (%positive cells/100) × no. transduced cells)/volume of virus (mL). After titration, cells were infected at MOI2 for 24 h and selected with puromycin. For sh*ALDH3B1* and negative control (GIPZ viral particle starter kit, Horizon Dharmacon), virus was directly purchased from the company (Horizon-Dharmacon) and used to infect the cells as described above. dCas9-*TET1*-CD-BFP contained the Blue Fluorescence Protein (BFP) and pgRNA contained the mCherry tag, GIC were then infected with MOI 1 of dCas9-*TET1*-CD-BFP construct plasmid overnight. BFP positive cells were FACS sorted and allowed to recover and expand before infection with MOI 1of individual or combination of single guides (pg or g1–6). GIC were then FACS sorted based on their co-expression of mCherry and BFP.

## Proliferation assay

Cells were plated (30 K) in 24 wells plates and detached after 3, 7, and 10 days and manually counted. Counting was repeated 3 times for each condition and for each time point, the experiment was performed twice. For proliferation assay in response to drug treatment, PGE1-OH (10 µM), Cyclotraxin-B (10 µM), DSF (1 µM), or vehicle were added to the media the day after the cells

were seeded. Media and drugs or vehicle were topped up every day and cells were harvested and counted at days 3, 7, and 10 and manually counted.

**Apoptosis**. Cells were plated (30 K) in 24 wells plates and treated 4 days with vehicle or drug (DSF 1 µM, Cyclotraxin-B 10 µM or PGE1-OH 10 µM), apoptosis was measured at end point following the manufacture's protocol (Caspase-3 assay kit, Abcam, ab39401). Briefly, cells were detached and resuspended in Cell Lysis Buffer for 10 min on ice, centrifuged and protein concentration was assessed from the supernatant. DTT was mixed with the reaction Buffer and added to each sample, then DEVD-pNA substrate was added to reach 200 µM final concentration and reaction mix was incubated 60−120 min at 37 °C. The absorbance of the chromophore p-nitroaniline (p-NA) formed by cleavage of the labelled substrate DEVD-pNA by the caspase-3 was read at 400–405 nm.

**Sphere-forming assay and limiting dilution assay**. Cells were plated (30 K) in 24 wells plates and treated 4 days with vehicle or drug (DSF 1 µM, CTX-B 10 µM or PGE1-OH 10 µM), then washed and detached with accutase and seeded ($1 \times 10^4$) in low adherent 12 well plates (costar) for 24 h for tumour sphere formation. The tumour spheres with diameters >20 µm were counted under a IRIS logos biosystem microscope (objective ×10). Results are expressed as the average of tumour sphere number on 10 pictures per well and standardized on the surface of one picture. Limiting dilution assay was performed following a published protocol[77]. Briefly cells were plated into low attachment 96-well plates with densities from 5 to 300 cells well in 100 µL of culture media. 20 µl of fresh media containing vehicle or drug (DSF, CTX-B, and PGE1_OH) were added every day for 10 days on GIC line of interest and one control line for each drug. Pictures were taken with aiRIS logos biosystem microscope (objective ×10).

**ALDH activity**. Cells were plated (30 K) in 24 wells plates and treated 4 days with vehicle or drug (DSF 1 µM, CTX-B 10 µM or PGE1-OH 10 µM) then processed following the manufacture's protocol (Aldefluor assay kit, Stem cell Technology). Cells were washed, detached with accutase, and $1 \times 10^6$ cells were resuspended in 1 ml Aldefluor assay buffer, half of them were transferred to a tube containing DEAB reagent (1.5 mM) (control tube) and the other half to an empty tube (test tube), then activated ALDFLUOR reagent was added to both tubes and the reaction was allowed to continue for 45 min at 37 °C. Cells were then kept on ice, centrifuged and pellets were resuspended in assay buffer. Flow cytometric analysis was conducted using a FACS LSRII (BD) and Aldefluor fluorescence was excited at 488 nm, and fluorescence emission was detected using a standard fluorescein isothiocyanate (FITC).

**Western blot**. Protein homogenates were obtained using RIPA buffer containing protease cocktail inhibitors (Santa Cruz Biotechnology). Protein concentration was determined by BCA assay (Pierce) and an equal amount (20 µg) was loaded into a 4–12% Bis-Tris precast gel (ThermoFisher Scientific). Proteins were separated by SDS-PAGE and blotted onto a nitrocellulose membrane (Whatman). Membranes were blocked with 5% non-fat dried milk (Santa Cruz)-0.1% TWEEN-TBS and incubated with the primary antibody diluted in blocking solution according to each antibody's recommendation: anti-ALDH3B1 antibody (Abcam) 1:500; anti-TrKB (Abcam) 1:500; anti-PTGER4 (Abcam) (Table S3). 1:500; anti-GAPDH (Sigma) 1:1000 and anti-Tubulin (Sigma) 1:5000. Membranes were then incubated with the peroxidase-conjugated mouse secondary antibody 1:5000 or rabbit secondary antibody 1:5000 (GE Healthcare, NA931V, and NA934V respectively) and visualized on a film or ChemiDoc using ECL kit (GE Healthcare). Quantification of protein expression was performed by densitometric analysis and normalized with ImageJ software.

**RT qPCR**. RNA was extracted from the cell pellet with Micro or Mini RNeasy kit (Qiagen 74104/74004) following the manufacturer's protocol. RNA was then treated with DNase I (Invitrogen) and 1µg was retrotranscribed by SuperScriptIII (Invitrogen, 18080093). 5 ng of cDNA template and SYBR Green primers (Table S3) were used to perform SYBR Green assay using SYBR Green PowerUp Master Mix (Applied Biosystems, A25742) run on a StepOne Real-Time PCR System (ThermoFisher). The housekeeping genes *GAPDH* or *ATP5B* were used.

**Statistical analysis for the wet lab experiments**. Sample processing was carried out blinded. Statistical analysis was performed using GraphPad software unless otherwise stated. Significance was determined with t-test, one-way ANOVA (with Sidak's test), or two-way ANOVA as appropriate, and displayed as the mean ± standard error (SEM). $p < 0.05$ was considered significant. Significance was indicated with asterisks: $*p < 0.05$; $**p < 0.01$; $***p < 0.001$. All variables were assumed to be normally distributed unless otherwise stated. Outliers were considered those data points furthest from the median value.

**Computational analysis (murine samples)**
*RRBS-Seq*. Reduced Representation Bisulfite Sequencing (RRBS) raw data for the reference samples were downloaded from the Omnibus GEO repository with the following accession numbers: GSE124532 for the Müller cells (3 replicas, Müller),

GSE81720 (4 replicas), and GSE78690 (3 replicas) for the hematopoietic primary stem cells (HSC), GSE60054 for the granulocyte-macrophage progenitor (2 replicas, GMP) and megakaryocyte erythroid progenitor cells (2 replicas, MEP), GSE86280 for the satellite skeletal muscle cells (6 replicas, Satellite). The reference samples were processed together with eNSC (3 replicas) and iNSC$^{Gibco}$ (3 replicas) from this study. Quality was assessed via FastQC (https://www.bioinformatics.babraham.ac.uk/). Trimming of low-quality reads and adapters was performed via TrimGalore v. 0.4.5-1 with "-rrbs" mode enabled (https://www.bioinformatics.babraham.ac.uk/). Subsequently, Bismark v. 0.22.1 was used for the alignment of reads, calling the subroutines "nucleotide-coverage" and "methylation-extractor" (https://www.bioinformatics.babraham.ac.uk/). The reference genome Ensembl GRCm38 was bisulfite-converted via "bismark_genome_preparation" prior to the alignment. Default settings were used, including the usage of Bowtie2 as a background aligner.

The coverage files were post-processed via edgeR[78], following the pipeline described in[79]. Briefly, CpG sites with a total coverage—i.e., the sum of methylated (Me) and unmethylated (Un) reads—greater than 10 across all samples were retained as well as only those in chromosomes 1–19 and X, resulting in 58,001 CpG sites shared across all samples. M-values were calculated as log2(Me+2)-log2(Un +2).

The annotation of regions was performed via ChIPseeker in R[80]. For this purpose, methylated and unmethylated sites were defined as having M-value > 0 and M-value < 0, respectively, and CpG sites falling within 1000 base pairs from the TSS were associated to gene promoters. Default categories were used and downstream regions were pooled with distal intergenic ones. The $\chi^2$ test and the effect size were used to statistically compare the patterns of annotated regions between iNSC$^{Gibco}$ and the other cell types, separately for methylated and unmethylated regions. The cutoff values for significance were $p < 0.05$ and effect size $> 0.1$.

*RNA-Seq*. Reference samples were downloaded from Omnibus GEO repository (GSE75300: Kang's NSC; GSE43916: Wapinski's NSC; GSE119741: Zhang's NSC; GSE64411: Friedmann-Morvinski's astrocytes and neurons; GSE75592: Schmid's astrocytes; GSE88982: Yanez's monocytes; please see the main text; for references see main text) and were processed together with the samples from this study (6 eNSC, 9 iNSC$^{Gibco}$ and 9 iNSC$^{N2B27}$): quality was assessed via FastQC and trimming of low-quality reads and adapters was performed via TrimGalore v. 0.4.5-1 with default parameters. The pseudoalignment package Salmon v. 0.13.1[81] was used to map all samples to a previously indexed reference genome Ensembl GRCm38 (release 90) to obtain transcript per million (TPM) expression levels, normalized for library size and transcript length. Transcript to gene quantification was performed via biomaRt in R and lowly expressed genes with TPM < 1 were filtered out. The expression values were finally z-transformed prior to plotting.

*Integrated RNA and RRBS analysis at gene level*. CpG sites with a total coverage greater than 10, across the 3 replicas of eNSC and iNSC$^{Gibco}$, were retained as well as only those in chromosomes 1–19 and X[79], resulting in 996,580 CpG sites for the comparison between the two cell types at gene level. Further, the function nearestTSS in the edgeR package was used to retain only the CpG sites in promoters, falling within 1000 base pairs from the gene TSS ($n = 419,763$). Reads were then pooled at gene level before computing M-values, resulting in 13,380 genes.

The integration of RNASeq TPM values led to a total of 11,107 genes with both expression and methylation status for the two cell types. Finally, median expression and methylation values across replicates were calculated for each gene.

*Principal component analysis and hierarchical clustering*. Principal component analysis, for both RRBS-Seq and RNA-Seq datasets, was performed with "prcomp" in R, and results were plotted with plotly (plotly.com). The first three components were retained, accounting for 90 and 87% of the total variance, for RRBS-Seq and RNA-Seq data, respectively.

Hierarchical clustering on the RNA-Seq data was performed in R using Euclidean distance and complete linkage. The R package dendextend was used for visualization and grouping of the samples, setting $k = 4$. The R package gplots were used to produce the heatmap in Fig. S2, showing the expression of top 500 genes across all samples. To improve visualization, the scaled log2-expression was plotted. The dendrogram was produced separately from the heatmap, with the normalized gene expression of the full list of genes instead of the top 500.

**Computational analysis (human samples)**
*RNA-Seq*. RNA-Seq data were processed using two separate pipelines. For hierarchical clustering and principal component analysis, transcript-level expression was estimated directly using the pseudoalignment package Salmon[81]. The results were then aggregated to obtain gene-level expression estimates in units of transcripts per million (TPM), which are normalized for library size and transcript length. For the purpose of identifying deregulated genes, gene counts were first estimated using the gapped alignment software STAR[82]. The reference genome used in both pipelines is Ensembl GRCh38 (release 90). DE genes are computed using the R package edgeR[78], on the basis of gene counts derived from STAR. Baseline dispersion is first estimated across all samples, before applying a two-sided quasi-likelihood F test to estimate genewise differential expression separately for

each patient, following the suggested best practices of the authors. P values are corrected for multiple hypothesis testing using the Benjamini-Hochberg correction, resulting in an estimated false discovery rate (FDR). We impose a minimum absolute fold change of 2 and a maximum FDR lower than 0.01.

*DNA methylation.* All DNA methylation data generated as part of this study were profiled using the Illumina HumanMethylation EPIC array. In addition, we included reference lines profiled on both the EPIC and Illumina HumanMethylation 450 K arrays. When comparing the two, EPIC array data were reduced to include only the same probes as the 450 K data (these have the same chemical design and are therefore comparable). Raw array data were first pre-processed using the ChAMP package in R to remove failed detections and probes with known design flaws[83], before normalization using the SWAN algorithm[84]. Differentially methylated regions are inferred following the strategy in the R package DMRForPairs[85]. Briefly, clusters are defined as groups of six or more probes separated by no more than 400 base pairs. Each cluster is then assessed for biological significance by requiring an absolute median change in the M value ($\Delta M$) greater than or equal to 1.4 between two comparison groups (e.g., iPSC vs ESC). Where replicates are present in either comparison group, we take the median value for each probe across the samples in the group before computing $\Delta M$. Where the cluster is deemed biologically significant, we assess statistical significance using the Mann−Whitney U test, comparing all probes in each condition. P values are adjusted for multiple hypothesis testing using the Benjamini−Hochberg FDR[86] and the cluster is declared a DMR if it has an FDR < 0.01.

The demethylation analysis for *PTGER4* via gRNA was carried out in a similar way: data were profiled using the Illumina HumanMethylation EPIC array, while the R ChAMP package was used to remove flawed probes and failed detections. Finally, the SWAN algorithm was used for normalization before comparing the results. *Classifying EPS lines:* The success of the reprogramming process was assessed using gene expression and DNA methylation data as follows. For DNA methylation, we compare our EPS lines and reference iPSC lines with two reference ESC lines (see main text for details of all references). For each EPS/iPSC line, we extracted core DMRs that were present in both ESC comparisons (i.e., the intersection of the two DMR lists). These were used to create the plots in Figs. S2e and 2f. For gene expression, we took an analogous approach, defining core DE genes as those present in both ESC comparisons. These were used to create the plot in Fig. S2h.

Computing GBM subgroup based on DNA methylation profile: Subgroups of GBM have previously been described based on analysis of methylome data[4]. Our bulk FFPE and GIC samples were assigned to subgroups as defined above on this basis of their methylation profiles using a published random forest classifier[18]. This accepts raw methylation array data as an input.

Integrative analysis of RNA-Seq and DNA methylation: DE and DMR results were combined with reference to the annotation for the Illumina HumanMethylation EPIC array used to obtain the DNA methylation results. In this annotation, a list of corresponding genes is provided for each CpG probe on the array together with the way in which the CpG probe is related to that gene (e.g., "falls within 1ˢᵗ exon"). A CpG probe is considered relevant to a gene if it lies anywhere within the unspliced gene (i.e., including intronic regions) or within 1.5 kb of the transcription start site. Each DMR comprises 6 or more CpG probes. In order to integrate DE and DMR results, we pool all related genes for these probes. A DE/DMR pair is considered concordant if the effect is in opposite directions (hypermethylated and downregulated/hypomethylated and upregulated).

We elected to focus on patient-specific gene targets identified as differentially expressed and methylated in just one patient GIC line relative to syngeneic iNSC. In order to prioritize this large list of genes, we imposed increasingly stringent requirements, defined in the main text. The lower tables in Fig. 4a show toy examples for four patients and four genes. Bold outlines indicate statistical significance. Gene A is not patient-specific at the level of DE or DM. Gene B has patient-specific DE but no corresponding DM. Gene C has patient-specific DE and is DM in multiple patients; this gene is on the longlist. Gene D has patient-specific DE and concordant patient-specific DM; this gene is on the shortlist.

*Computing enriched pathways in DE gene sets.* The Ingenuity Pathway Analysis (IPA) platform was used to compute enriched gene sets in DE gene lists. All pathways with an uncorrected p value < 0.05 were retained, though stricter thresholds were applied in subsequent analyses. The genes involved in enriched pathways were exported separately from IPA in order to generate network visualizations and to check for cross-talk between IPA signatures and the cell type signatures used in xCell.

In order to compare the extent of pathway enrichment between syngeneic DE gene lists and DE gene lists arising from a comparison of our GIC with a reference NSC line (Fig. S5b), we retained all pathways with p < 0.005. For each pathway, we computed the number of patients with a DE list enriched for that pathway (GIC relative to syngeneic iNSC, H9 NSC or Gibco NSC). The resulting vectors were compared using the Wilcoxon signed-rank test in order to determine whether the number of pathways captured with each comparator differed significantly. The rank correlation metric was used to determine DE lists captured more pathways (i.e., the effect direction). We found that the syngeneic DE lists captured

significantly more pathways than using the H9 ($p = 0.029$) or Gibco ($p = 0.003$) reference comparators. A similar but more significant difference is obtained if the sum of –log10(p) values across patients is used instead of the binary yes/no metric described here.

Network analysis: Cytoscape v. 3.7.2[87] was used to perform network analysis on the (significantly) enriched pathways (uncorrected p value < 0.01) found with IPA (QIAGEN), in the 10 syngeneic GIC-iNSC transcriptomic comparisons. Enrichment map and Autoannotate were used for visualization and clustering, respectively. The edge similarity threshold was set at 0.5. Various layout modifications were implemented, such as edge bundling.

Inferring cell-type proportions in bulk tissue: Gene expression quantification (TPM values) of bulk GBM FFPE samples was processed using the xCell web application[40] in order to infer the proportions of 64 immune and stroma cell types within each bulk sample. This inference is based on a curated set of one or more gene signatures corresponding to each cell type, numbering 489 in total. For the purpose of eliminating cross-talk between IPA pathway signatures and xCell, we also downloaded these signatures.

Correlating cell type proportions with pathway enrichment: Before analyzing the correlation between inferred cell-type proportions and IPA pathway enrichment results, it was first necessary to exclude combinations for which the cell type signature overlaps extensively with the pathway. Therefore, any occurrence of an IPA pathway for which >10% of the genes were also contained within a cell type signature was ignored. For each cell type reported by xCell and pathway reported by IPA, we computed the Spearman rank correlation and associated exact p value across all patients between the –log10(p) values from the pathway enrichment and the inferred cell type proportion. As this correlation is rank-based, there is no need to correct pathway enrichment p values for multiple hypothesis testing. An example of two such computations is shown in Fig. S7a.

DNA methylation and RNA expression data from the independent GIC cohort HGCC, publicly available, were used to assess the proportion of patients with *PTGER4*, *NTRK2*, and *ALDH3B1* signatures in a wider population of patients. For the differential gene expression analysis of GICs from the HGCC cohort, microarray data that had been pre-processed (normalized and Combat batch adjusted) was provided by our collaborator Professor Sven Nelander, Uppsala University. Differential gene expression analysis was then performed in an R workspace using the RStudio environment and limma package 83 (version 3.40.6). DNA methylation was processed with Illumina array, as described above, βvalue were used for further analysis. Briefly, for each of the three genes, average RNA sequencing and DNA methylation value were calculated and patients with high methylation and low expression for *PTGER4* were identified while patients with low methylation and high expression were selected for *NTRK2* and *ALDH3B1*.

Schematics were created with BioRender.com

**Reporting summary**. Further information on research design is available in the Nature Research Reporting Summary linked to this article.

## Data availability

The authors declare that all data supporting the findings of this study are available within the article, its supplementary information files, and the public repositories as stated below. The datasets generated in this study are available as raw data in the NCBI Gene Expression Omnibus database with SuperSeries number GSE155994, which includes all mouse data GSE154367, the human transcriptomic GSE154958, and the human methylation data GSE155985. Source data are provided with this paper.

Publicly available datasets used in the study: GSE124532, GSE81720, GSE78690, GSE60054, GSE86280, GSE75300, GSE43916, GSE119741, GSE64411, GSE75592, https://www.bioinformatics.babraham.ac.uk/. Source data are provided with this paper.

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

## Acknowledgements
This work is funded by grants from Brain Tumour Research (Centre of Excellence award to S.M.), Cancer Research UK (C23985/A29199 programme award to S.M.), Barts Charity (MGU0447 programme grant to S.M.), The Brain Tumour Charity (GN-000389 clinical research training fellowship to T.M.), NIHR (CL-2019-19-001 to TM) and The Willoughby Fund Trustees (studentship to AAD). Part of the study was funded by the National Institute for Health Research to UCLH Biomedical research centre (BRC399/NS/RB/101410 to S.B.). S.B. is also supported by the Department of Health's NIHR Biomedical Research Centre's funding scheme. J.L.T. and C.L.R.M. are supported by the Engineering and Physical Sciences Research Council (EP/N006615/1 programme grant) and F.M.B by The British Heart Foundation (BHF RG/14/2/30616). We thank the staff of our animal facility for help with the daily care of our mouse colony. We thank Dan Pennington for advice on ethical requirements for blood collection for research use and Reiss Browning for performing tail vein injections. We acknowledge the use of data generated by the TCGA Research Network: https://www.cancer.gov/tcga.

## Author contributions
C.V., L.G., J.R.B., M.C., A.A.D. and X.Z. designed and performed experiments and analyzed results. G.R. and N.P. designed and performed all computational analyses. J.R, T.E. and N.A. identified and consented the patients for the study. S.B. obtained ethical approval and supervised human sample collection and pathological and molecular analysis of the biopsies. Y.M.L. performed image analysis. T.O.M. assisted with intracranial-injection experiments. J.V., S.Na. and F.M.B. co-designed and supervised the Tregs experiments. J.L.T. and C.L.R.M. performed the GAG experiments. T.J. assisted with karyotyping. V.R., S.N., D.S., Y.Y.L. and P.L. shared datasets and essential expertize. S.M. conceived and supervised the study. C.V., G.R. and S.M. wrote the manuscript with contribution from all authors.

## Competing interests
The authors declare no competing interests.
