## [Peer Review File · Nature Communications]

Reviewers' Comments:

Reviewer #1:

Remarks to the Author:

The authors present the validation of a model using patient matched GIC and iNSC that they carefully characterize by their methylome and transcriptome. Further, the validity of the model of patient derived iNSC is validated using respective mouse samples for which endogenous NSC are readily available for comparison. These experiments show the comparability between iNSC and endogenous NSCs, and indicate some differences when comparing eNSCs and iNSCs coming from different mouse strains. This paves the way for the authors to support their hypothesis of the individual make-up of iNSC, and the rationale for using patient matched iNSCs for determining personalized treatment strategies. Subsequently the data is used for target identification for known drugs, and prediction of patient specific responsiveness based on the molecular make-up. Several hits are further studied as proof of principle for the specificity of molecular make-up and corresponding/predicted drug response or resistance. One feature predicts the infiltration of Tregs, others comprise targets and respective drugs, of which some have been previously established and reported from GSC-based studies. The study uses state of the art technologies that is well described and referenced.

Overall this is a very interesting study that reads well. Beside the fact that the model(s) can be used as a tool for drug development, it provides insights into individual differences of normal and tumor that is of scientific interest and clinically relevant for drug response. The authors suggest that the pipeline could be developed for individualized patient management.

Comments/Questions:

- The identified targets could be assessed in the TCGA data, evaluating functional methylation (negative correlation of DNA methylation with corresponding gene expression, for silencing; or the opposite, loss of methylation and overexpression). This would provide the reader with information on the size of the patient population that may potentially respond to a proposed treatment.
- Disulfiram has been assessed previously for treatment effects in GBM models (BTIC ~ GIC) (Lun et al 2016, DOI: 10.1158/1078-0432.CCR-15-1798) including testing in clinical trials as cited. Therefore, the presented results are basically a rediscovery, validating the approach. Furthermore, the efficacy of the single agent as shown seems not really impressive in the mouse model (Fig 6E). This is also reflected in the fact that the drug has been tested in a combination therapy in patients. It could be of interest here, to focus on the specificity of the treatment effect relevant for biomarker development for patient selection.
- The authors propose that the outlined strategy could be developed into a pipeline for individualized patient management. What are the time lines for patient matched predictions? How could generalization help with timely predictions for individual patients. How would potential targets/drug pairs be prioritized (size effect)? Further, single agent therapies have been very disappointing and current strategies move to rational combination therapies, how could this be integrated into the proposed pipeline.

The study may profit from biostatistical review.

Minor comments :

- In the context of disulfiram treatment, it is not clear why the authors discuss the effect of disulfiram on MGMT in absence of TMZ treatment in their experiments. Could be removed.
- increase font size of legends within graphics for readability throughout, many are not readable (e.g. Fig 4B is not understandable as nothing can be read and the graphics are too small to understand what is represented.
- there is no Figure 4D

Monika Hegi

Reviewer #2:

Remarks to the Author:

Vinel et al. present a very comprehensive study using various technological and biological state of the art technologies. Thus, given the overall high innovative character of the assay developed, the work surely shall received high priority for publication.

Some points to consider:

1. The main revolutionizing concept is the capacity of biologically validate computational-predicted pharmacological interventions, scoring therapeutic effectivity and off-target risk, in personalized pathophysiological relevant cell models. The toxicity aspect address the main bottle neck that arose when testing anti-cancer stem cell directed therapies in clinical trials: off-target effects to "non tumor stem cells" when applied systemically. The developed technology will be more disruptive if

a) providing evidences by showing negative control experiment. In other words, select a drug that is predicted to target GIC and INSC similarly and conduct biological validation. Two drugs and in vitro experiments is sufficient.

b) gastro-intestinal toxicity – as the main problematic effect in clinical anti-stem cell oncology trials: The authors must include the limitation of their procedures to not being able to predict off-target risk of predicted & (in neuro-context) off-target validated drugs on gastro-intestine tissue. Or is there a chance to include data on this aspect?

2. More details and discussion on feasibility of realistic implementation in clinical setting/ dissemination potential of described technology: How much extra effort for surgeon to collect fibroblast tissue? Minimal requirements on dura piece (size, time out of the brain until in vitro processing). Maybe share some suggestions how to make it feasible for the clinical stakeholder to participate in such program, or is it simply interest/motivation of the individuals itself, that makes such a cooperation possible? What is the success rate of growing primary meningeal fibroblasts, what its the percentage of successful generation stem cells (what is the efficacy percentage), from these fibroblasts? Please provide a sketch what is the time line of the ideal experimental procedure: time of resection – primary model – neural induction – expansion - functional 3D in vitro assay (confirmed with biological replications) . Is this achievable in 6 month after resection, the time window most of our patients would strongest benefit from individualized pharmacotherapy. Alternatively, would you speculate that, in case of recurrence (almost 100% the case), once can directly start with treatment based on primary tumor in silico prediction results? Is this approach – especially the context of using iNSC to reduce off-target effects of drugs - feasible / transferrable to the sector of personalized pharmaco- management of cerebral metastasis. Chemotherapy, even in non-tailored to the individual patient, is strongly underrepresented/appreciated in the clinical management of non-classical cerebral metastasis (mamma or lung primary tumor), that occur unfrequently, and would certainly benefit from such risk-assessed suggestions for interventions.

Minor:

Please invent a name /abbreviation for this assay to improve dissemination/referral for others to this assay.

Quantification of pluripotency markers if human iNSCs available?

-Fig1 D and E: the figure legends are to small for printed versions. The PDF needs to be viewed at 200% to read the graphic

Response to reviewers' comments

Reviewer #1 (Remarks to the Author):

The authors present the validation of a model using patient matched GIC and iNSC that they carefully characterize by their methylome and transcriptome. Further, the validity of the model of patient derived iNSC is validated using respective mouse samples for which endogenous NSC are readily available for comparison. These experiments show the comparability between iNSC and endogenous NSCs, and indicate some differences when comparing eNSCs and iNSCs coming from different mouse strains. This paves the way for the authors to support their hypothesis of the individual make-up of iNSC, and the rationale for using patient matched iNSCs for determining personalized treatment strategies. Subsequently the data is used for target identification for known drugs, and prediction of patient specific responsiveness based on the molecular make-up. Several hits are further studied as proof of principle for the specificity of molecular make-up and corresponding/predicted drug response or resistance. One feature predicts the infiltration of Tregs, others comprise targets and respective drugs, of which some have been previously established and reported from GSC-based studies. The study uses state of the art technologies that is well described and referenced.

Overall this is a very interesting study that reads well. Beside the fact that the model(s) can be used as a tool for drug development, it provides insights into individual differences of normal and tumor that is of scientific interest and clinically relevant for drug response. The authors suggest that the pipeline could be developed for individualized patient management.

Comments/Questions:

- The identified targets could be assessed in the TCGA data, evaluating functional methylation (negative correlation of DNA methylation with corresponding gene expression, for silencing; or the opposite, loss of methylation and overexpression). This would provide the reader with information on the size of the patient population that may potentially respond to a proposed treatment.*

We agree that gathering a wider estimation of the patients displaying functional methylation of the identified target genes would be of interest. We have attempted to interrogate the TCGA data, although we realised that the number of patients for whom both RNA and DNA methylation datasets were available was rather small and the data were from bulk tumours. However, we took advantage of the Human Glioma Cell Cultures (HGCC) database (<https://www.hgcc.se/>), a cohort of 71 GIC with both RNA and DNA methylation datasets.

Firstly, we established the average of RNA expression and methylation for each of the 3 genes: PTGER4, NTRK2 and ALDH3B1. Then we selected patients with methylation above the average ("high methylation") together with an expression below the average ("low expression") for PTGER4. The patients with this signature ("High meth/Low expr") represented 34% of the cohort. For NTRK2, patients with "low methylation" and "high expression" represented 21% and for ALDH3B1, 22.5% of GIC derived from patients of the cohort showed the signature "Low meth/High expr". These data raise the possibility that a substantial proportion of GBM patients could be amenable to treatment with these compounds, provided the DM/DE signature is confirmed at syngeneic level.

These results have been added to the MS, page 11 and Fig. S8C.

- Disulfiram has been assessed previously for treatment effects in GBM models (BTIC ~ GIC) (Lun et al*

2016, DOI: 10.1158/1078-0432.CCR-15-1798) including testing in clinical trials as cited. Therefore, the presented results are basically a rediscovery, validating the approach.

This is correct. However, we would like to highlight that the cited study focussed on the effect of disulfiram on GBM exploiting its property to chelate Copper ion (Cu) leading to inhibition of proteasome activity, as had been previously shown in melanoma patients. Moreover, because DSF is a potent and direct inhibitor of MGMT, this study attempted to address a potential combinatorial effect of TMZ and DSF + Cu in unmethylated GBM patients. Interestingly 14% of the patients enrolled in the study responded, which could be encouraging bearing in mind that they had not been selected on the basis of the “Low meth/High expr” ALDH3B1 signature we describe.

Our syngeneic comparison study adds a novel classification parameter: Hypomethylation and upregulation of ALDH3B1 as a signature predicting response to ALDH targeting therapy such as Disulfiram. The differential vulnerability of GIC with and without the signature we observed, was independent of the MGMT methylation status. It could be of interest to re-assess the trial results on the basis of our signature, if appropriate datasets are available.

Furthermore, the efficacy of the single agent as shown seems not really impressive in the mouse model (Fig 6E). This is also reflected in the fact that the drug has been tested in a combination therapy in patients. It could be of interest here, to focus on the specificity of the treatment effect relevant for biomarker development for patient selection.

We are completely in agreement that a combination treatment would be very interesting and potentially more potent and we will certainly explore this in future experiments. Of note, the analysis of the HGCC GIC collection described above has highlighted that 8/71 lines showed a signature for 2 out of 3 genes of interest in this paper, thus 11.3% could be predicted to respond to a combined therapy targeting either PTGER4/NTRK2 (4%) or NTRK2/ALDH3B1 (7%), should these data be confirmed at syngeneic level. Moreover, combination of these drugs with standard of care therapeutic approaches could also be explored. The importance of combination therapy and how this assay could contribute to its design is now mentioned in the manuscript, page 19.

• The authors propose that the outlined strategy could be developed into a pipeline for individualized patient management. What are the times lines for patient matched predictions? How could generalization help with timely predictions for individual patients. How would potential targets/drug pairs be prioritized (size effect)? Further, single agent therapies have been very disappointing and current strategies move to rational combination therapies, how could this be integrated into the proposed pipeline.

The purpose of the current study is to provide proof of principle that patient-specific actionable targets can be identified with the novel GIC/syngeneic iNSC platform (SYNGN) we have set up. At present it takes approximately 6 months between surgery and derivation of GIC and fibroblasts, reprogramming of the fibroblasts to EPSC, differentiation to iNSC, expression/DNA methylation profiling and analysis with drug response prediction. This is not yet ideal for implementation in an adjuvant patient setting. However, we have secured funding for and are working on improving this timeline and are confident it can be brought down to 3-4 months by taking advantages of reprogramming methods now applicable to white blood cells, hence bypassing the need to establish fibroblasts culture as well as automation of experimental steps and taking advantage of established sequencing and analytical approaches.

Prioritization of targets/drug pairs will be predominantly based on drug availability/FDA approval, optimal ADMET (absorption, distribution, metabolism, excretion, toxicity), blood-brain-barrier penetrance, potency and selectivity. We are currently assessing suitability of the approach for rematching at recurrence, which would benefit from the iNSC already been available, hence would

be significantly faster and we are indeed trialling its suitability for prediction of combination approaches. Results of these experiments are however, beyond the scope of this proof of concept study.

The study may profit from biostatistical review.

Biostatistical review has been carried out throughout the MS.

Minor comments :

- In the context of disulfiram treatment, it is not clear why the authors discuss the effect of disulfiram on MGMT in absence of TMZ treatment in their experiments.

This sentence has been removed from the manuscript.

-increase font size of legends within graphics for readability throughout, many are not readable (e.g. Fig 4B is not understandable as nothing can be read and the graphics are too small to understand what is represented.

-there is no Figure 4D .

Font sizes have been reviewed and where appropriate increased throughout all figures.

Monika Hegi

Reviewer #2 (Remarks to the Author):

Vinel et al. present a very comprehensive study using various technological and biological state of the art technologies. Thus, given the overall high innovative character of the assay developed, the work surely shall received high priority for publication.

Some points to consider:

1. The main revolutionizing concept is the capacity of biologically validate computational-predicted pharmacological interventions, scoring therapeutic effectivity and off-target risk, in personalized pathophysiological relevant cell models. The toxicity aspect address the main bottle neck that arose when testing anti-cancer stem cell directed therapies in clinical trials: off-target effects to “non tumor stem cells” when applied systemically. The developed technology will be more disruptive if a) providing evidences by showing negative control experiment. In other words, select a drug that is predicted to target GIC and INSC similarly and conduct biological validation. Two drugs and in vitro experiments is sufficient.

This is an important point which we are happy to address experimentally.

As a first negative control, we would like to highlight that whilst patient 18 showed specific downregulation (Fig S9E) and hypermethylation of PTGER4 gene in the GIC compared to the iNSC (Fig 4B), hence the prediction that GIC would be sensitive to the agonist PGE1-OH, which we

validated; in patient 30 PTGER4 was expressed at low levels in both GIC and iNSC (Fig S9E) and was not differentially methylated (Table S3) and thus both are predicted to be sensitive to PTGER4 agonist, which we have confirmed experimentally (Fig 5A bottom). Moreover, PTGER4 is expressed at high levels in both GIC and iNSC in patient 26 (Fig 4B) and is not differentially methylated (Table S3), hence the prediction would be that they are both resistant to PGE-OH1 and we now show the experimental validation of this prediction (MS page 12 and Fig S9F).

We feel that these are probably the strongest negative controls, as the analytical algorithm we used is not designed to identify non responding drug matches

However, we have manually identified the Hexokinase 2 (HK2) gene to be not DE and not DMR in patient 61 and highly expressed in both GIC and iNSC of this patient (Table S3) with both lines predicted to be sensitive to an inhibitor. There is no drug predicted to interact with this gene in DGIdb, however Ikarugamycin was recently described to inhibit HK2 in pancreatic cancer cells [1]. We have now confirmed the sensitivity of both GIC and iNSC derived from patient 16 to this drug (Fig. 1A this rebuttal).

Moreover, the gene TUBB6 encoding for the Tubulin Beta 6 Class V is specifically hypermethylated and downregulated in the GIC derived from patient 18 compared to iNSC (Table S3) and consequently iNSC (TUBB6 upregulated and hypomethylated) are predicted to be sensitive to an inhibitor of this gene such as the Colchicine (Table S3) with the GIC predicted to be more resistant. Our experimental data show that iNSC are indeed more sensitive to colchicine (Fig. 1B this rebuttal).

Fig. 1: Drug treatment on GIC (black curves) and iNSC (grey curves) of patient 61 (top) and 18 (bottom) with a range of doses from 2nM to 10μM for Ikarugamycin (light green dots) and 1nM to 2.5μM of Colchicine (burgundy dots). Results are expressed as a percentage of viability on the vehicle and were measured at end point, area under the curve (AUC) was calculated from the percentages of viability (n=3 experimental repetitions, mean ±SEM, two tailed t test. *p≤0.05, **p≤0.01, ***p≤0.001).

b) gastro-intestinal toxicity – as the main problematic effect in clinical anti-stem cell oncology trials: The authors must include the limitation of their procedures to not being able to predict off-target risk of predicted & (in neuro-context) off-target validated drugs on gastro-intestine tissue. Or is there a chance to include data on this aspect?

This is an important point, which we are happy to address. Drug-induced gastrointestinal toxicities (DIGTs) are undoubtedly a challenge in anti-stem cell oncology trials. Off-target effects difficult to predict with *in vitro* assays, which have been shown to be of limited use or animal models, which are low throughput and only partially recapitulating human physiology, are well known [2]. 3D models of

gastrointestinal tract are emerging as a promising tool and have been previously used to assess DIGTs [2] [3]. Hence, we reasoned that we could take advantage of the availability of syngeneic EPSC lines to generate gastro-intestinal organoids of the patients of our cohort predicted to respond to the drugs identified in our pipeline. We show that EPSC-derived intestinal organoids from patients 18, 30 and 19 can be generated and do not show overt evidence of DIGT, as assessed by Ki67 (proliferation) and cCasp3 (apoptosis) staining, when treated with the drug concentrations effective in the GIC (page 15 and 18, Fig. S16). Whilst it is of course still to be proven that this assay accurately predicts the response in patients, it is encouraging that this approach is viable at pre-clinical level. Importantly, we now show that assessment of the drug impact on proliferation and apoptosis in SYNGLICO captures the effect observed in 2D cultures and in vivo (Fig. 5H/S11A and Fig. 6F and G).

2. More details and discussion on feasibility of realistic implementation in clinical setting/ dissemination potential of described technology:

How much extra effort for surgeon to collect fibroblast tissue? Minimal requirements on dura piece (size, time out of the brain until in vitro processing). Maybe share some suggestions how to make it feasible for the clinical stakeholder to participate in such program, or is it simply interest/motivation of the individuals itself, that makes such a cooperation possible? What is the success rate of growing primary meningeal fibroblasts, What its the percentage of successful generation stem cells (what is the efficacy percentage), from these fibroblasts?

The success rate of establishing a fibroblast culture is very high (92% in our hands) and only a thin strip of dura mater (from the surgical incision) is required (1-2cm X 0.1cm), hence no major extra effort is required from the surgical team. The efficacy of our reprogramming method is also very high (88%) and induction of iNSC from EPSC is even higher (93%).

We would like to highlight though, that the work presented in this paper is a proof of concept that the novel pipeline we have established is suitable for prediction of drug response in a proportion of GBM patients at pre-clinical level. We are very keen to explore how this can be taken forward to the clinic and we have secured funding to do so but results of these efforts are very much beyond the scope of this manuscript. At present it takes approximately 6 months between surgery and derivation of GIC and fibroblasts, reprogramming of the fibroblasts to EPSC, differentiation to iNSC, expression/DNA methylation profiling and analysis with drug response prediction. This is not yet ideal for implementation in an adjuvant patient setting. However, the timeline could be brought down to 3-4 months by taking advantages of reprogramming methods now applicable to white blood cells, hence bypassing the need to establish fibroblasts culture as well as automation of experimental steps and taking advantage of established sequencing and analytical approaches. We are confident the assay stands a definite chance to progress to the clinic in due course.

Please provide a sketch what is the time line of the ideal experimental procedure:

time of resection – primary model – neural induction – expansion - functional 3D in vitro assay (confirmed with biological replications) . Is this achievable in 6 month after resection, the time window most of our patients would strongest benefit from individualized pharmacotherapy.

Alternatively, would you speculate that, in case of recurrence (almost 100% the case), once can directly start with treatment based on primary tumor in silico prediction results?

A sketch of the pipeline is now provided (Fig. 8). On the applicability of the pipeline at clinical level, see response to the point above. Moreover, we are pursuing two additional lines of research: as this

reviewer mentions the possibility to apply the prediction on the primary GIC/iNSC comparison at recurrence is being assessed, as is the possibility to derive GIC from the recurrence and use the already established iNSC for a new faster prediction, given the availability of the syngeneic comparator. Although, these future developments are beyond the scope of this manuscript, they are now mentioned in the manuscript, page 19.

Is this approach – especially the context of using iNSC to reduce off-target effects of drugs - feasible / transferrable to the sector of personalized pharmaco- management of cerebral metastasis. Chemotherapy, even in non-tailored to the individual patient, is strongly underrepresented/appreciated in the clinical management of non-classical cerebral metastasis (mamma or lung primary tumor), that occur unfrequently, and would certainly benefit from such risk-assessed suggestions for interventions.

This is an interesting point, which could certainly be explored. Availability of syngeneic EPSC which allows to take into account off target effects at various cellular levels (iNSC, intestinal organoids etc.) could be powerful also for chemotherapy planning of cerebral metastasis.

Minor:

Please invent a name /abbreviation for this assay to improve dissemination/referral for others to this assay.

We are proposing to call the approach “SYNGN assay”(syngeneic comparison of GIC and iNSC).

Quantification of pluripotency markers if human iNSCs available?

Pluripotency markers (NANOG, POU5F1 and KLF4) are downregulated in iNSC as compared to EPSC, relevant data from RNA sequencing have been added to the manuscript (FigS2K).

-Fig1 D and E: the figure legends are too small for printed versions. The PDF needs to be viewed at 200% to read the graphic

Font sizes have been reviewed and where appropriate increased throughout all figures.

Two in vivo experiments, which were being performed at the time of the first submission have now been completed. They validate important aspects of the work presented in this manuscript and have been added to this revised version of the paper: GAGs-mediated Tregs attraction in vivo (page 10 and Fig. 3G, Fig. S7J) and pathogenic relevance of the methylation at the PTGER4 promoter in vivo (page 14, Fig. 5E).

References

1. Jiang, S.H., et al., *Ikarugamycin inhibits pancreatic cancer cell glycolysis by targeting hexokinase 2*. FASEB J, 2020. **34**(3): p. 3943-3955.
2. Nag, D., et al., *Auranofin Protects Intestine against Radiation Injury by Modulating p53/p21 Pathway and Radiosensitizes Human Colon Tumor*. Clin Cancer Res, 2019. **25**(15): p. 4791-4807.

-
-
3. Peters, M.F., et al., *Human 3D Gastrointestinal Microtissue Barrier Function As a Predictor of Drug-Induced Diarrhea*. *Toxicol Sci*, 2019. **168**(1): p. 3-17.

Reviewers' Comments:

Reviewer #1:

Remarks to the Author:

The authors have answered to all questions to my satisfaction, and this interesting manuscript is now ready for publication.

Reviewer #2:

Remarks to the Author:

The authors performed a good job to answer my concerns. I believe SYNGN will be influential for the field. I suggest paper to be accepted in present form.